# Optogenetic gamma stimulation rescues memory impairments in an Alzheimer's disease mouse model

Guillaume Etter [1], Suzanne van der Veldt[1], Frédéric Manseau[1], Iman Zarrinkoub[1], Emilie Trillaud-Doppia[1] & Sylvain Williams[1]*

Slow gamma oscillations (30–60 Hz) correlate with retrieval of spatial memory. Altered slow gamma oscillations have been observed in Alzheimer's disease. Here, we use the J20-APP AD mouse model that displays spatial memory loss as well as reduced slow gamma amplitude and phase-amplitude coupling to theta oscillations phase. To restore gamma oscillations in the hippocampus, we used optogenetics to activate medial septal parvalbumin neurons at different frequencies. We show that optogenetic stimulation of parvalbumin neurons at 40 Hz (but not 80 Hz) restores hippocampal slow gamma oscillations amplitude, and phase-amplitude coupling of the J20 AD mouse model. Restoration of slow gamma oscillations during retrieval rescued spatial memory in mice despite significant plaque deposition. These results support the role of slow gamma oscillations in memory and suggest that optogenetic stimulation of medial septal parvalbumin neurons at 40 Hz could provide a novel strategy for treating memory deficits in AD.

---

[1] McGill University & Douglas Mental Health University Institute, Montreal, Canada. *email: sylvain.williams@mcgill.ca

In the hippocampus, rhythmic activity occurs within distinct frequency bands that include theta (θ, 4–12 Hz) and gamma (γ, 30–120 Hz) frequency oscillations. Recently, slow (30–60 Hz) and fast (60–120 Hz) gamma oscillations have been proposed to have functionally distinct roles in memory function[1] (see Colgin for review[2–4]). Slow gamma oscillations recorded in CA1 originate from the stratum radiatum (rad) and are coupled to slow gamma oscillations in CA3[5], a hippocampal subfield essential for memory retrieval[6,7]. On the other hand, fast gamma oscillations have been suggested to originate from the stratum lacunosum-moleculare (lm) and are coupled to fast gamma oscillations in the medial entorhinal cortex[5,8]. Slow gamma oscillations have been proposed to support memory retrieval as neuronal activity in the hippocampus can predict exploration of familiar spatial trajectories during epochs of slow gamma oscillations[1,9]. In spite of evidence suggesting the role of fast and slow gamma oscillations in memory encoding and retrieval, respectively, causal evidence is lacking, in part because of the technical challenges associated with specifically controlling gamma oscillations. In this context, recent advances in the development of optogenetics have allowed the control of neuronal activity at high frequencies and it has been shown that parvalbumin (PV) neurons from the basal forebrain can pace cortical gamma oscillations at ~ 40 Hz when stimulated at that frequency[10]. Previous neuroanatomical work has shown that PV cells located in the medial septum (MSPV cells) send direct projections to GABAergic interneurons in the hippocampus, giving rise to septohippocampal feedforward inhibitory control[11]. Among the GABAergic cells receiving inputs from MSPV cells, hippocampal PV interneurons are instrumental in pacing hippocampal oscillations, at least in the theta frequency band[12]. Supporting this, optogenetic stimulation of MSPV cell projections in the hippocampus are associated with direct, frequency specific pacing of hippocampal oscillations[13,14]. Although PV interneurons in the hippocampus were shown to be part of a microcircuit regulating memory consolidation[15,16], the exact role of MSPV cells in memory encoding and retrieval remains poorly described. Previous reports show that spatial coding remains intact in pyramidal cells during theta oscillation entrainment in spite of a reduction in place cell firing frequency[14], but the exact effects of such stimulation in memory tasks has not been addressed yet.

Oscillatory activity in the gamma range is altered in Alzheimer's disease (AD) patients[17,18] and in AD mouse models[19–21] (see Mably for review[22]). The mechanisms behind such alterations are not known, but recent evidence suggests that either soluble amyloid beta (Aβ) or its fibrillary forms may affect synaptic and neuronal functions[23,24], and would probably contribute to numerous network alterations in rodents[25] long before plaques can be detected. In fact, defects of synaptic function and cognition have frequently been reported before any neurodegeneration or plaque deposition can be observed in the brain of transgenic mice overexpressing the amyloid precursor protein (APP)[26–28], in triple transgenic mice (3xTg)[29], as well as in wild-type mice injected with Aβ[30,31]. Although the coupling of gamma oscillations to theta phase has frequently been reported to support memory processing[32–35] and relies on fast-spiking hippocampal PV interneurons[36,37], decreased theta–gamma phase-amplitude coupling has been reported in vitro for an APP mouse model[19] as well as in vivo in 3xTg AD mice[20], suggesting that altered phase-amplitude coupling could play a key role in AD pathophysiology. Chronic optogenetic stimulation of cortical parvalbumin neurons in AD mouse models was recently shown to reduce levels of Aβ[38] and was associated with improved recognition memory[39]. On the other hand, others have restored both gamma oscillations and normal cognition when upregulating interneuron activity in J20-APP mice[40]. Although this suggests that slow gamma oscillations may support memory retrieval, and that decrease in gamma oscillations as well as their coupling to theta could be at the origin of memory impairments in AD, the exact contribution of slow gamma oscillation to memory retrieval remains poorly understood. In the current study, we show that APP overexpression leads to spatial memory deficits and that specifically increasing slow gamma oscillations during memory retrieval using optogenetic stimulation is sufficient to ameliorate memory performance in an AD mouse model.

## Results

**Hippocampal gamma oscillations are altered in PVJ20+ mice.** In order to examine changes of theta and gamma rhythms in a mouse model that recapitulates some of the hallmarks of AD and allow for optogenetic control of hippocampal oscillations, we generated a mouse line that was obtained by crossing J20 mice overexpressing APP with PV-Cre mice. We termed this new mouse line PVJ20 (Fig. 1a), and compared offsprings that expressed PV-Cre combined with APP overexpression (PVJ20+) to littermates expressing PV-Cre without APP overexpression (PVJ20−). PVJ20+ mice overexpressed APP leading to plaques in the hippocampal formation (supplementary fig. 1a–g) and surrounding cortices at 6 months of age (supplementary fig. 1h–k, supplementary table 1). PVJ20− mice displayed robust spatial memory deficits as measured in the appetitive version of the Barnes maze task (supplementary fig. 2a–d), and failed to alternate successfully in the delayed non-match-to-place task (supplementary fig. 2e–f). Considering that theta and gamma oscillations are highly state dependent and that both increase in power and frequency during running[41], we first examined dynamic locomotor-related electrophysiological changes of CA1 hippocampal theta and gamma oscillations during wakefulness, whereas mice freely explored a circular platform (Fig. 1b). In a subset of mice, we implanted silicon probes in PVJ20− and PVJ20+ mice (supplementary fig. 3a) and confirmed their location with histology (supplementary fig. 3b), in addition to ripple power being maximal in the stratum pyramidale (supplementary fig. 3c), and theta phase reversal between the stratum rad and stratum lm (supplementary fig. 3d, e). We measured power spectral densities (PSD) and found reduced slow (30–60 Hz) gamma in PVJ20+ mice (2ANOVA, $F_{(1, 56)} = 9.89$, $p = 0.0027$ for the main effect of genotype; $n = 6$), whereas fast (60–120 Hz) gamma was not significantly different (2ANOVA, $F_{(1,56)} = 0.8096$, $p = 0.3721$ for the main effect of genotype; $n = 6$; supplementary fig. 3f). Comodulograms that represent the coupling strength of gamma (30–120 Hz) amplitudes to theta (4–12 Hz) phases also reveal decreased slow gamma (but not fast gamma) amplitude coupling to theta in the stratum lm in two example mice (supplementary fig. 3h; only theta frequencies with maximum coupling of gamma are preserved in this representation. See supplementary fig. 3g for detailed comodulograms). Consequently, we next focused our analyses in larger groups of mice in the stratum lm using single microelectrodes (Fig. 1c). It is noteworthy that while slow gamma oscillations have been suggested to originate from stratum rad using ICA decomposition[5], they can be recorded in stratum lm where theta oscillation amplitudes are the strongest. We took advantage of this feature when normalizing gamma oscillations to theta power, which gave more consistent results. As expected, wavelet spectrograms that resolve both time and frequency of oscillatory signals revealed sustained theta (4–12 Hz) oscillations during active exploration but intermittent gamma oscillations in PVJ20− mice (Fig. 1d, left panel). In PVJ20+ mice, gamma oscillations were decreased in power compared with control PVJ20− littermates (Fig. 1d, right panel),

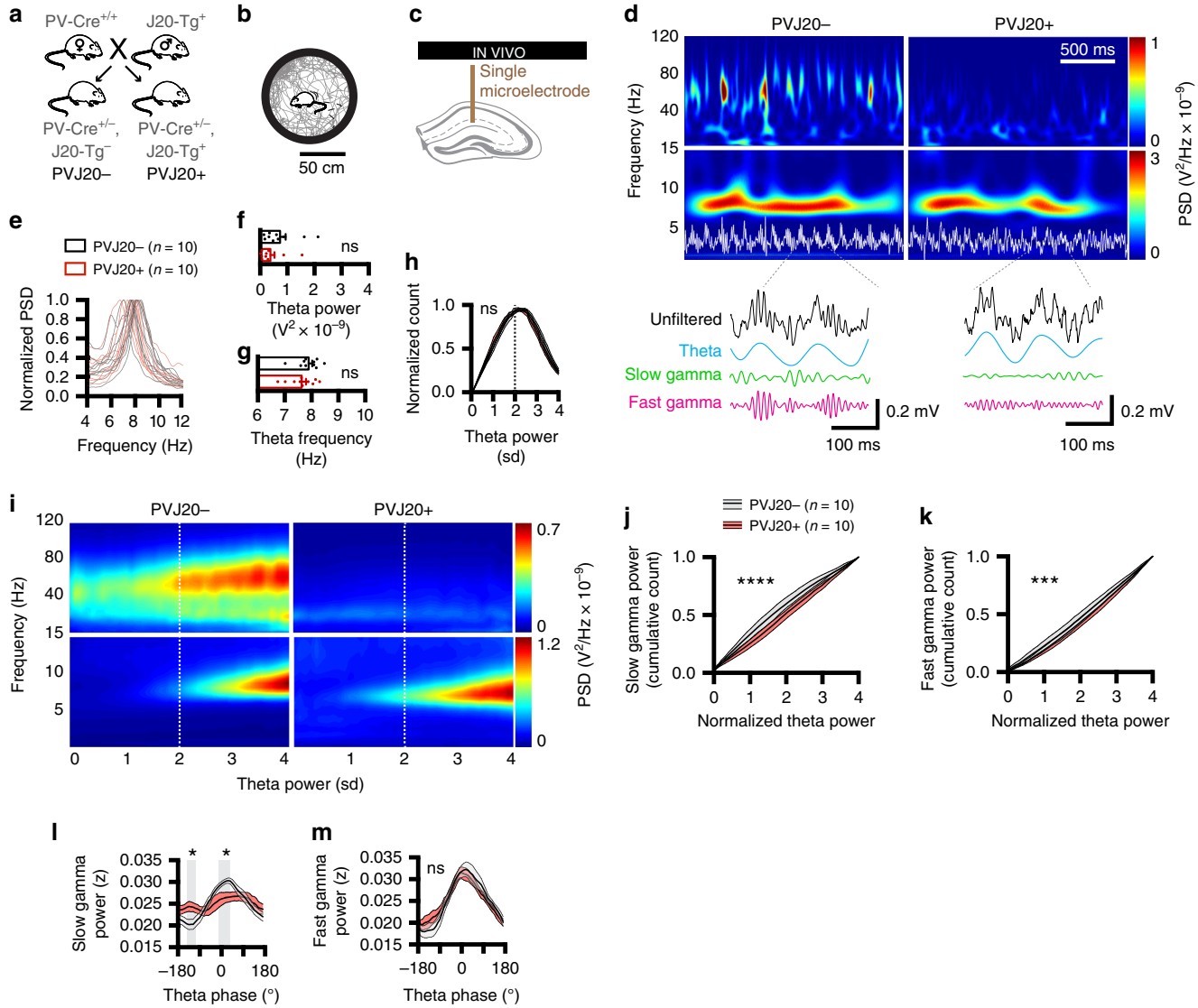

**Fig. 1** Altered gamma oscillations in PVJ20 mice. **a** PVJ20 mouse line breeding scheme. **b** Exploration trajectory on a circular platform. **c** Recording configuration. **d** 2 s example spectrograms for PVJ20− (left) and PVJ20+ (right) mice recorded in CA1-lm while running on the circular plateform. **e** Normalized power spectra in the theta band for each individual PVJ20+ (red) and PVJ20− (black) littermates. **f** Theta power from raw LFP. **g** Theta peak frequency. **h** Normalized histogram counts of normalized theta power. **i** Example spectrograms sorted by normalized theta power from CA1-LFP recorded in PVJ20− (left) and PVJ20+ (right) mice. **j** cumulative distribution of slow gamma oscillations. **k** same for fast gamma. **l** phase-amplitude coupling of slow gamma power over theta phases. **m** Same for fast gamma. $n = 10$ PVJ20+ mice and $n = 10$ PVJ20− mice were used in panels **e**–**h**, **j**–**m**. Color codes in **d** and **i**: blue, low power; red, high power. Data expressed as mean ± SEM. ns, not significant; *, $p < 0.05$; ***, $p < 0.001$; ****, $p < 0.0001$. Test used in **f**, **g**: independent student $t$ test. **h**, **j**–**m** 2ANOVA, Sidak's multiple comparisons post hoc test). Source data are provided as a Source Data file.

whereas theta oscillations in PVJ20+ mice remained unchanged compared to control PVJ20− littermates (Fig. 1e) both in terms of power (unpaired $t$ test, $t_{18} = 1.437$, $p = 0.1678$; Fig. 1f) as well as frequency (unpaired $t$ test, $t_{18} = 1.278$, $p = 0.2175$; $n = 10$ PVJ20− mice and $n = 10$ PVJ20+ mice; Fig. 1g). Because gamma oscillations are transient, whereas theta oscillations are more consistent and comparable between PVJ20− and PVJ20+ mice, we next sorted wavelet convoluted CA1 local field potential (LFP) power in the gamma band and normalized this data to theta power (Fig. 1i). This method allowed us to compare similar behavioral states in both groups. We found that PVJ20+ mice displayed decreased slow gamma (2ANOVA, $F_{(1, 720)} = 44.12$, $P < 0.0001$ for main effect of genotype; Fig. 1j) as well as fast gamma (2ANOVA, $F_{(1, 720)} = 13.07$, $p = 0.0003$; $n = 10$ PVJ20− mice and $n = 10$ PVJ20+ mice; Fig. 1k) power compared with PVJ20− mice. Since gamma oscillations are modulated by the

phase of theta oscillations, we also sorted gamma power by theta phase (only considering theta states, where normalized theta exceeded 2 sd) and found that PVJ20+ mice displayed decreased phase-amplitude coupling of slow gamma oscillations compared with PVJ20− littermates (2ANOVA, $F_{(39, 720)} = 2.488$, $p < 0.0001$ for interaction between theta phase and slow gamma power, Sidak's post hoc test; $n = 10$ PVJ20− mice and $n = 10$ PVJ20+ mice; Fig. 1l). Interestingly, we found that fast gamma coupling to theta was not altered in PVJ20+ mice (2ANOVA, $F_{(39, 720)} = 0.6321$, $p = 0.9618$ for interaction between theta phase and genotype; $n = 10$ PVJ20− mice and $n = 10$ PVJ20+ mice; Fig. 1m).

**Optogenetic control of MSPV cells in PVJ20+ mice.** Optogenetic stimulation of MSPV neurons[14] or their projections to the hippocampus[13] are associated with frequency specific pacing of

hippocampal oscillations. Owing to the widespread innervation of MSPV neurons in the hippocampus (see supplementary fig. 4c), targeting these neurons directly in the medial septum could provide a powerful and extensive control of hippocampal oscillations. To determine whether hippocampal oscillations could be experimentally generated in PVJ20+ mice, we expressed the ultrafast optogenetic actuator ChETA in MSPV cells of PVJ20+ mice (supplementary fig. 4a, b) as this construct allows better control of neuronal spiking at high stimulation frequencies[42]. On average, $96.84 \pm 1.39\%$ (mean ± SEM) of eYFP cells were parvalbumin-positive (PV+), whereas $86.23 \pm 2.45\%$ of PV+ cells co-expressed eYFP ($n = 15$ mice). Expression of eYFP was not found in cholinergic cells (supplementary fig. 4d, e, g). We then determined the accuracy of optogenetic control of MSPV neurons in medial septum slices in vitro (Fig. 2a). MSPV cells were identified by their eYFP expression (Fig. 2b) and their relative lack of frequency adaptation (Fig. 2c). As described previously[43,44], these parvalbumin neurons could fire at high frequencies in response to increased injected current (Fig. 2d; supplementary fig. 5b). We then determined optimal illumination duration and intensity for 40 and 80 Hz (which are respectively within slow and fast gamma frequency bands), and thus analyzed responses of MSPV neurons to 40 Hz, 12 ms pulses, and 80 Hz, 6 ms pulses at various laser intensities (Fig. 2e–h; supplementary fig. 5). MSPV cells could be entrained at 40 Hz (Fig. 2e) and 80 Hz (Fig. 2f). We found that both 6 and 12 ms light pulses induced large photocurrents in MSPV cells (RM-2ANOVA, $F_{(1, 4)} = 17.13$, $p = 0.0144$), whereas photocurrents were not significantly different between the two pulse widths (RM-2ANOVA, $F_{(1, 4)} = 3.905$, $p = 0.1193$; Fig. 2g). In addition, stimulation at both 40 and 80 Hz induced robust responses in MSPV cells (2ANOVA, $F_{(1, 8)} = 12.05$, $p = 0.0084$ for the main effect of stimulation wavelength) and did not differ significantly in terms of ability to elicit spikes at 10 mW (2ANOVA, $F_{(1, 8)} = 1.109$, $p = 0.3231$; Fig. 2h). We then analyzed hippocampal CA1-LFP responses to MSPV optogenetic stimulation in vivo using the parameters found in MSPV neurons in vitro (Fig. 2i). We determined that optimal responses could be obtained with laser power between 10 and 20 mW (supplementary fig. 6). To generate gamma oscillations in PVJ20+ mice, we compared the effects of 10 Hz (theta band), 40 Hz (slow gamma band), and 80 Hz (fast gamma band) optogenetic stimulation and found that 40 Hz stimulation were most effective at increasing slow gamma power in the hippocampus (1ANOVA, $F_{(3, 30)} = 6.202$, $p = 0.0021$; Fig. 2l). Importantly, the power of theta oscillations was not altered by either 40 or 80 Hz optogenetic stimulation (Fig. 2k). Fast gamma band oscillations could not be significantly increased by 80 Hz stimulation (Fig. 3m), which could be owing to the fact that this frequency band is much wider than slow gamma or theta, and that higher frequencies are generally much lower in amplitude than lower frequencies. Nevertheless, this stimulation did induce a significant increase of 80 Hz oscillations as measured by the PSD at that frequency ($3.63 \pm 0.99$ fold baseline PSD, one sample $t$ test, $t_{(6)} = 2.633$, $p = 0.0389$; Fig. 2n). We also tested theta-burst stimulation, but this paradigm did not significantly increase slow or fast gamma power (supplementary fig. 7). In one example mouse, we also tested various stimulation frequencies and found frequency specific responses ranging from 5 to 80 Hz (supplementary fig. 8).

**MSPV optogenetic stimulation rescues gamma oscillations**. To further assess the effects of the optogenetic stimulation, we used both intra- (638 nm laser stimulation; see supplementary figs. 5 and 6) as well as inter-individual (YFP) controls (Fig. 3b) and performed alternating 2 s periods of 40 Hz stimulation and 2 s without stimulation. This paradigm increased the number of

stimulation trials and allowed to sample responses across more behavioral states. We measured the power of slow gamma oscillations over theta phases (Fig. 3c, top row), corresponding stim/baseline ratio (Fig. 3c, bottom row), as well as group averages (Fig. 3d–g). We found that these 40 Hz stimulation increased slow gamma phase-amplitude coupling (2ANOVA, $F_{(1, 6)} = 17.93$, $p = 0.0055$ for the main effect of injected construct; Fig. 3d). As expected, 80 Hz optogenetic stimulation did not increase slow gamma phase-amplitude coupling (Fig. 3f). Interestingly, fast gamma phase-amplitude coupling was not increased by 80 Hz stimulation (2ANOVA, $F_{(1, 240)} = 1.617$, $p = 0.2048$; Fig. 3g) which could be owing to the fact that PVJ20+ mice do not display altered fast gamma phase-amplitude coupling to theta (Fig. 1m). In contrast, 40 Hz stimulation increased both slow gamma power (2ANOVA, $F_{(1, 12)} = 18.63$, $p = 0.0010$; Fig. 3h) as well as theta–gamma coupling strength measured using the modulation index[45] (MI; 2ANOVA, $F_{(1, 12)} = 12.20$, $p = 0.0044$; Fig. 3f), suggesting that this stimulation pattern could be employed to efficiently restore hippocampal slow gamma oscillations in PVJ20+ mice. Importantly, these effects were not dependent on the phase detection method as both Hilbert transform and a broad-band, peak-to-peak detection method[46] provided comparable phase-amplitude coupling responses (supplementary fig. 9). It is noteworthy that during stimulation, there was a significant increase in slow gamma power in PVJ20+ mice (RM-2ANOVA, $F_{(1, 120)} = 317.0$, $p < 0.0001$), and when compared with PVJ20− littermates, they did not differ significantly in terms of slow gamma phase-amplitude coupling to theta (2ANOVA, $F_{(1, 480)} = 1.231$, $p = 0.2678$), suggesting that 40 Hz stimulation can restore gamma activity to normal levels. In one example mouse, we performed optogenetic stimulation while recording CSD signals from a silicon probe spanning CA1 transversal axis (supplementary fig. 10a, b). Both baseline CSD amplitudes and optogenetic responses were highest in the *lm* regardless of the stimulation pattern used, whereas only 40 Hz stimulation also elicited significant response in the stratum pyramidale (supplementary fig. 10h).

**40 Hz optogenetic stimulation rescues memory in PVJ20+ mice**. Finally, to examine whether decreased slow gamma oscillations may underlie memory defects in AD conditions, we determined whether 40 Hz medial septum stimulation could improve memory performance in PVJ20+ mice. Although older studies suggest that the encoding process of episodic memories may be impaired in AD conditions[47–49], more recent work indicates that memory retrieval is dysfunctional in AD conditions[22,50]. To address this question, we used a novel object place recognition task (Fig. 4a) that discriminates between encoding (sample) and retrieval (test) phases. Importantly, PVJ20 mice did not display any significant difference in thigmotaxis or total exploration time in this task (supplementary fig. 11). A group of PVJ20+ implanted with single electrodes in CA1-lm (Fig. 4b) recapitulated decreased slow gamma power (2ANOVA, $F_{(1, 240)} = 33.61$, $p < 0.0001$; Fig. 4c) and fast gamma power (2ANOVA, $F_{(1, 240)} = 239.8$, $p < 0.0001$; Fig. 4d), as well as slow gamma phase-amplitude coupling (2ANOVA, $F_{(39, 240)} = 1.547$, $p = 0.0264$ for interaction between theta phase and slow gamma power; Fig. 4e) compared with PVJ20− littermates during the test phase of this task. Here again, fast gamma phase-amplitude coupling was not found to be different (2ANOVA, $F_{(39, 240)} = 0.1547$; Fig. 4f). In PVJ20+ mice transfected with ChETA (Fig. 4g), we were able to induce an increase in slow gamma oscillations power during the task (Fig. 4h) and such increase could be sustained throughout the 10 min of test (RM-1ANOVA, $F_{(1.243, 3.728)} = 1.529$, $p = 0.3022$ for the main effect of time; Fig. 4i, j). Importantly, neither theta

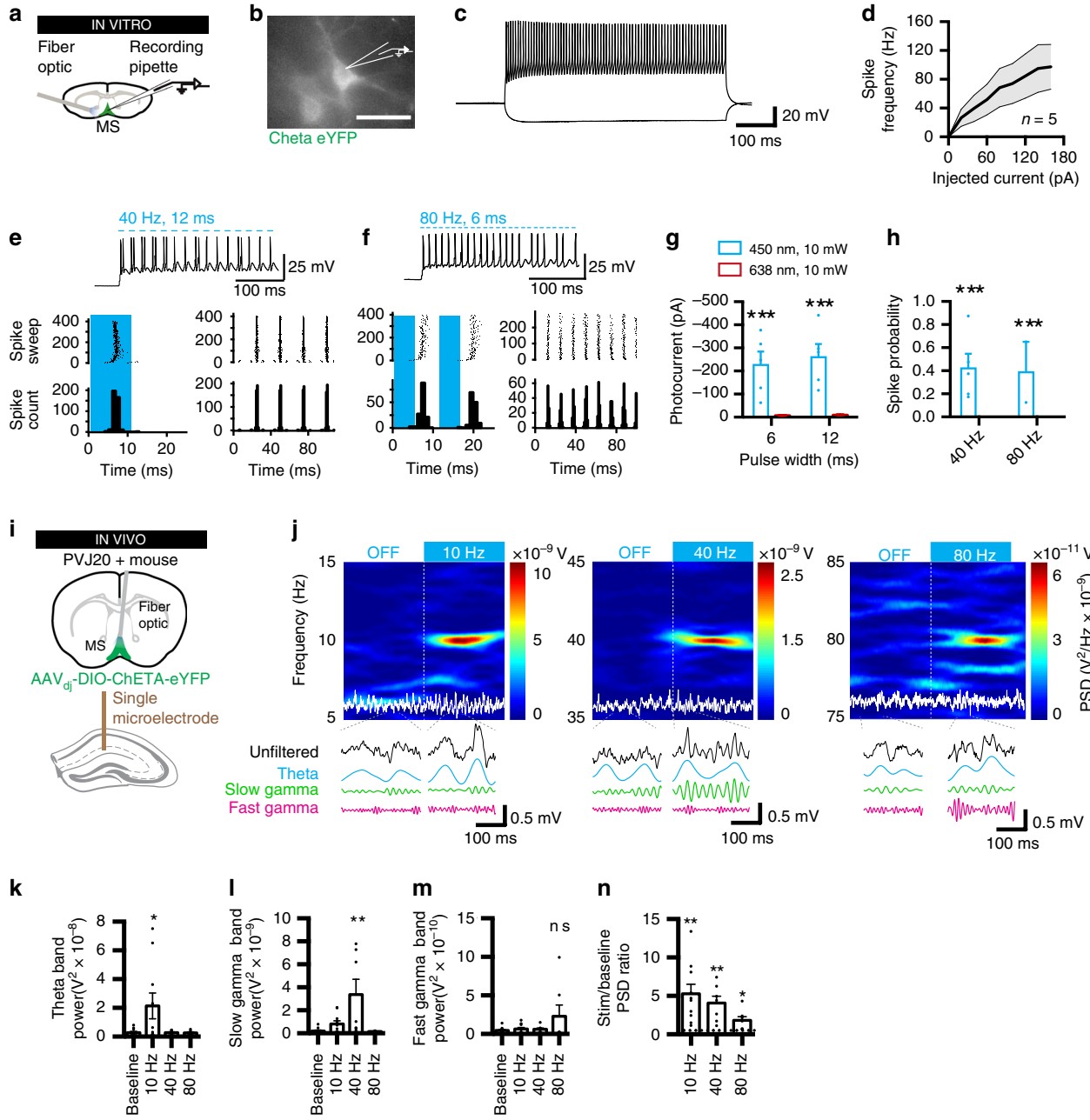

**Fig. 2** Optogenetic control of medial septum parvalbumin cells can increase gamma oscillations in PVJ20+ mice. **a** In vitro recording configuration. **b** Example eYFP-ChETA + MSPV neuron visually identified. Scale bar: 50 μm. **c** Fast-spiking response of MSPV cell in response to injected current in current clamp. **d** Average spike frequency of identified eYFP-ChETA + MSPV neurons in response to current injection. **e** Example trace of neuron in response to 40 Hz, 50% duty-cycle light stimulation (top) and associated spike timing histogram in response to each pulse (bottom). **f** Same for 80 Hz, 50% duty-cycle stimulation. **g** Photocurrents in response to either 450 or 638 nm (control), 6 or 12 ms, 10 mW laser stimulation. **h** Spike probability for 40 and 80 Hz laser stimulation. $n = 5$ neurons are used in **d**. **i**, In vivo recording configuration. **j** Spectrogram before and during 10, 40, or 80 Hz optogenetic stimulation with raw traces (white) supperimposed. Effects of optogentic stimulation on theta **k**, slow gamma **l**, and fast gamma **m** band power. **n** Ratio of stimulation/baseline PSD for each stimulation frequency. In **k–n**, $n = 10$ PVJ20+, ChETA mice were used to test 10 Hz stimulation, $n = 7$ PVJ20+, ChETA mice were used to test 40 Hz stimulation, and $n = 7$ PVJ20+, ChETA mice were used to test 80 Hz stimulation. Color codes for **j**: blue, low power; red, high power. Data expressed as mean ± SEM. ns, not significant. *, $p < 0.05$; **, $p < 0.01$; ***, $p < 0.001$. Test used in **g**, **h**: 2 ANOVA, Sidak's multiple comparisons post hoc test; **k–n**: 1ANOVA, Holm–Sidak's multiple comparisons post hoc test. Source data are provided as a Source Data file.

frequency (Wilcoxon matched-pairs signed rank test, $p = 0.8750$; Fig. 4k) nor phase distribution (RM-2ANOVA, $F_{(1, 120)} = 7.080 \times 10^{-013}$, $p > 0.9999$; Fig. 4l) were altered during such prolonged stimulation. Mice only implanted with fiber optic implants were then subjected to the novel object place recognition task (Fig. 4m). Remarkably, the discrimination index was higher in PVJ20− YFP control mice ($0.61 \pm 0.02$) and PVJ20 + ChETA

mice stimulated at 40 Hz during the test phase ($0.66 \pm 0.05$) compared with PVJ20 + YFP mice ($0.47 \pm 0.06$; RM-2ANOVA, $F_{(4, 32)} = 2.813$, $p = 0.0416$; Sidak's post hoc test; partial $\eta^2 = 0.26$; Fig. 4n). To determine whether the effects of the 40 Hz stimulation were specific for the retrieval phase, we also employed a group of PVJ20 + ChETA mice that were only stimulated at 40 Hz during the sample phase of the task. Although the

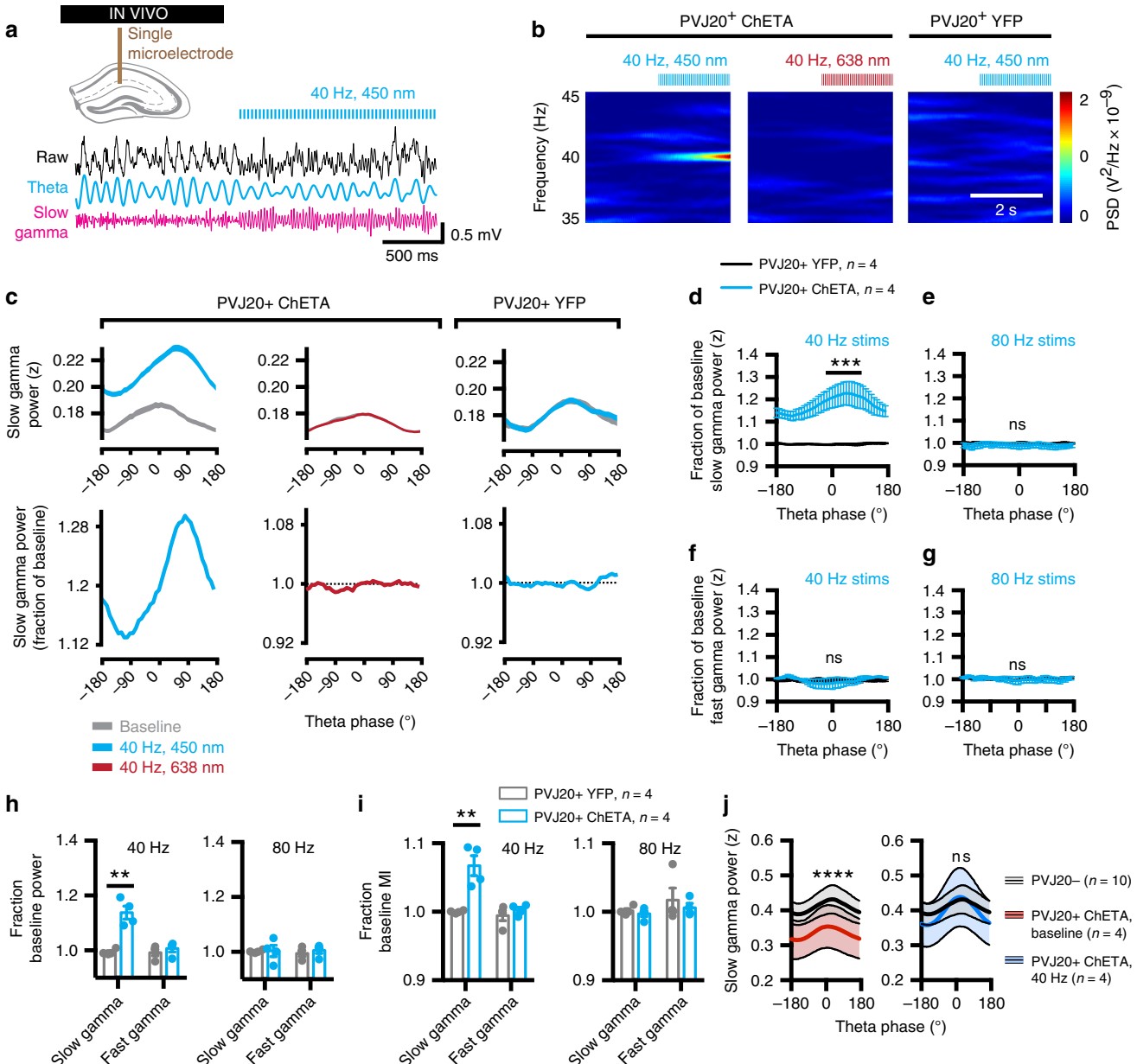

**Fig. 3** In 40 Hz stimulations of medial septum parvalbumin neurons increase slow gamma power and coupling to theta oscillations. **a** Recording configuration and representative example of CA1-LFP before and during 40 Hz stimulation at 450 nm. Raw LFP (black), theta band filtered trace (blue), and gamma band filtered trace (pink). **b** Representative example Fourier spectrograms of CA1-LFP before and during either 40 Hz, 450 nm stimulation in one PVJ20+ mouse transfected with ChETA (left), or 40 Hz, 638 nm stimulations in the same mouse (center), or 40 Hz, 450 nm stimulation in a different PVJ20+ mouse transfected with a control, YFP construct (right). **c** 40 Hz stimulation at 450 nm increase gamma power in ChETA transfected PVJ20+ mice (left; 2ANOVA, $F_{(1, 72960)} = 5393$, $P < 0.0001$ for the main effect of stimulation, $n = 913$ stimulation) but not in a PVJ20+ YFP control (right; $n = 1689$ stimulation) or when using 638 nm laser stimulation (center; $n = 918$ stimulations). Data is expressed in z scored gamma power (top) or stim/baseline ratio (bottom), thickness of lines represents SEM. **d**, **e** Group averages of stim/baseline slow gamma power ratio. **f**, **g** Group averages of stim/baseline fast gamma power ratio. **h** Group averages of stim/baseline slow and fast gamma power for 40 and 80 Hz. **i** Same for modulation index (MI). **j** Slow gamma amplitude coupling to theta phase before (left) or during (right) 40 Hz optogenetic stimulation. $n = 4$ PVJ20+, ChETA mice and $n = 4$ PVJ20+, YFP mice were used in **d–j**. Color codes in **b**: blue, low power; red, high power. Data expressed as mean ± SEM. ns, not significant. **, $p < 0.01$; ***, $p < 0.001$; ****, $p < 0.0001$. Test used in **d–j** 2ANOVA, Sidak's multiple comparison post hoc test. Source data are provided as a Source Data file.

discrimination index (0.59 ± 0.06) of that group did not significantly differ from PVJ20 + YFP mice (0.47 ± 0.06), they also did not significantly differ from PVJ20 + ChETA mice stimulated during the retrieval phase (Sidak's multiple comparison test; Fig. 4n). Although fast gamma oscillation coupling to theta is not altered in PVJ20+ mice (Fig. 1s, supplementary fig. 3f), 80 Hz stimulation can pace MSPV cells at that frequency (Fig. 2f-h, j, n,

supplementary figs. 6h, 8c, 10j). Therefore, to control for the frequency specificity of optogenetic stimulation, we used a group of PVJ20 + ChETA mice that received 80 Hz optogenetic stimulation during the test phase of the task. Mice stimulated at 80 Hz during the retrieval test did not perform significantly better than PVJ20 + YFP mice (0.55 ± 0.05) and did not significantly differ from PVJ20 + 40 Hz mice stimulated during the retrieval phase.

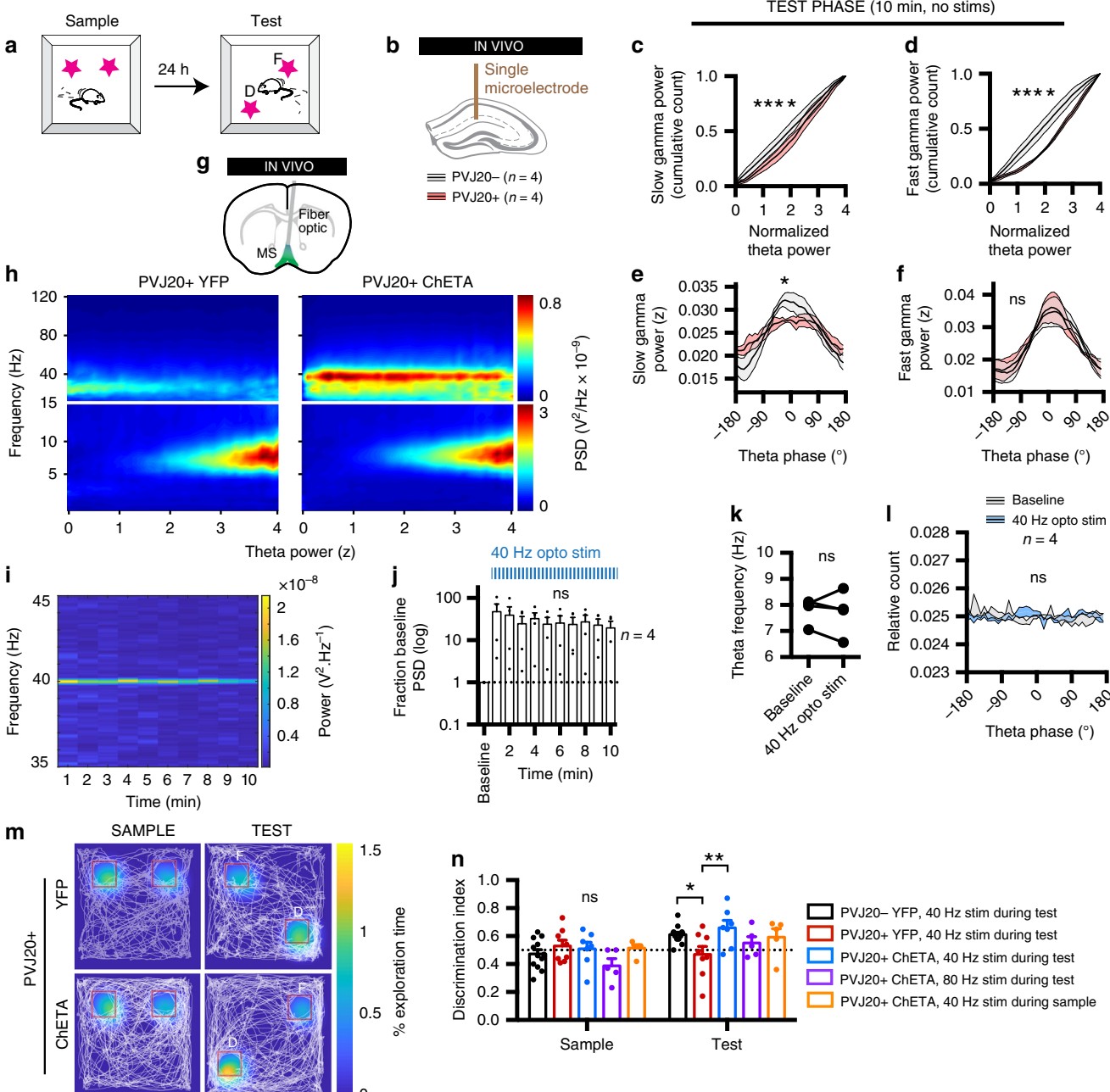

**Fig. 4** 40 Hz optogenetic stimulation of medial septum parvalbumin neurons rescues spatial recognition. **a** Novel object place recognition paradigm. F, familiar object. D, displaced object. **b** Recording configuration. PVJ20 + and PVJ20− control littermates were recorded in CA1-lm during the test phase (no stimulation). **c** Cumulative distribution of slow gamma oscillations power. **d** Same for fast gamma oscillations. **e** Phase-amplitude coupling of slow gamma power over theta phases. **f** Same for fast gamma. **g** PVJ20+ mice were transfected with either the ChETA or a YFP construct and were implanted with a fiber optic over the medial septum. **h** Example spectrograms sorted by normalized theta power during 40 Hz medial septum parvalbumin stimulation for 10 min (blue, low power; red, high power). **i** Time-resolved spectrogram using 1 min time bins (blue, low power; yellow, high power). **j** PSD at 40 Hz was measured over time and normalized by a baseline recorded in absence of stimulation. **k** Theta frequency before and during 40 Hz optogenetic stimulation. **l** Distribution of theta phase before and during 40 Hz optogenetic stimulation. **m** Example exploration (white trace) overlayed with total relative nose exploration heatmap (blue, no exploration; yellow, maximum exploration time). **n** Discrimination index in the sample and test phases. $n = 4$ PVJ20+ mice and $n = 4$ PVJ20− mice were used in panels **c–f**, **j–l**. **n**, $n = 12$ PVJ20−, YFP mice, $n = 8$ PVJ20 +, YFP mice, and $n = 7$ PVJ20 +, ChETA mice were stimulated at 40 Hz during the test phase. $n = 5$ PVJ20+, ChETA mice were stimulated at 80 Hz during the test phase, and $n = 5$ PVJ20+, ChETA mice were stimulated at 40 Hz during the sample phase. Data expressed as mean ± SEM. *, $p < 0.05$; **, $p < 0.01$; ****, $p < 0.0001$. Test used in **c–f** and **l**: 2ANOVA, Sidak's multiple comparisons post hoc test. **j** RM-1ANOVA. **k** Wilcoxon matched-pairs signed rank test. **n** RM-2ANOVA, Sidak's multiple comparison post hoc test. Source data are provided as a Source Data file.

## Discussion

We found that PVJ20+ mice display decreased gamma oscillations and that specifically reinstating slow gamma oscillations during memory retrieval using optogenetics was sufficient to reverse memory loss. Although previous work has described the effects of chronic optogenetic stimulation[39], we show here that memory improvements can be obtained using acute optogenetic treatment. Importantly, we have separated analyses of gamma power and phase-amplitude coupling and have found that both slow and fast gamma power were reduced at comparable theta amplitudes, results that are comparable with those previously found in both J20-APP[40] and taupathy[51] mouse models. We also observed that only the coupling of slow gamma amplitude to theta phases was reduced, with fast gamma coupling remaining intact, which was also found in a 3xTg mouse model[20]. We have summarized in supplementary table 2 our behavioral data including the effects of optogenetic treatment on hippocampal physiology.

Although some studies have focused on decreasing plaque load to improve brain function in APP mice[38,39,52,53], there is a large body of evidence showing that Aβ deposits do not correlate significantly with AD progression in humans[54–61]. In that respect, our results support recent findings that memory can be improved in the absence of Aβ clearance by applying optogenetic stimulation specifically during memory retrieval[50,62]. In contrast to previous studies suggesting that AD alters encoding functions[47–49], or reconsolidation[63], our results are in agreement with AD memory dysfunction being related to defects in memory retrieval[50,62]. Stimulation during retrieval was sufficient to restore memory performance, which suggests that encoding processes are maintained in PVJ20 mice. Although stimulating during encoding did not significantly increase memory performance, we did observe higher scores in some of the mice tested. Further in-depth behavioral investigation of acute optogenetic stimulation could shed light on such memory improvements.

Considering that MSPV neurons project directly onto GABAergic interneurons in the hippocampus[11], it is likely that our optogenetic stimulation modulates the excitability of hippocampal pyramidal neurons to some extent. Although release of feedforward inhibition would be expected, previous studies have described reduction of place cell peak firing frequency during such stimulation, at least when stimulating MSPV interneurons in the theta band[14]. Interestingly, we found that in vitro, MSPV cells display a wide range of maximum firing frequencies, as shown previously[43,44]. This could explain in part the discrepancies in our ability to control their activity for high frequencies at the single cell level in vitro, whereas we were successful in pacing hippocampal rhythms up to 80 Hz in vivo. Importantly, our optogenetic gamma stimulation did not alter theta power, frequency, or phase distribution and did not alter locomotion or anxiety behavior in behaving mice. Based on our histological and in vivo results, we hypothesize that our optogenetic stimulation can realistically only control a portion of the MSPV cell population. Moreover, despite the absence of ChETA transfection in cholinergic cells in our conditions, indirect cholinergic effects cannot be entirely excluded since ~ 10% of GABAergic project onto cholinergic cells within the medial septum[64]. Although we found that MSPV interneurons are silent in vitro, these cells were previously found to be frequency-locked to theta oscillations during theta states[65], and locked to varying theta phases[66]. Considering our findings in vivo, it is likely that these cells preserve their phase locking to theta while being entrained at higher (40, 80 Hz) frequencies. Further studies employing dedicated single unit recordings of MSPV cells during optogenetic stimulation should be performed to confirm that this is indeed the case. Furthermore, although using single electrodes has allowed us to increase the

statistical power of our results, these recording conditions do not permit independent component analysis (ICA) or other decomposition methods that would allow us to identify gamma generators within the hippocampus, and the associated effects of optogenetic stimulation on them. Therefore, while our optogenetic stimulation could generate artificial slow and fast gamma rhythms in hippocampal layers that normally do not exhibit such oscillations, our stimulation might either be associated with increased natural gamma oscillations, or a superimposition of natural and artificial gamma oscillations.

Treatments for AD that target Aβ have not been successful so far. In part, this is probably owing to the fact that synaptic alterations occur long before plaques can be detected[67,68]. Our study is relevant in the context of AD treatment as we found that acute optogenetic stimulation is sufficient to control gamma oscillations directly and reinstate memory recall. Our results suggest that more direct neural network manipulations could be employed as novel therapeutics to reverse memory loss in AD, and supports the role of hippocampal slow gamma oscillations in spatial memory retrieval.

## Methods

**Animals**. All procedures were approved by the McGill University Animal Care Committee and the Canadian Council on Animal Care. Male mice expressing a mutant form of the human amyloid protein precursor bearing both the Swedish (K670N/M671L) and the Indiana (V717F) mutations (APPSwInd) under the control of the PDGFB promoter [B6. Cg-Tg(PDGFB-APPSwInd)[28], (Jackson laboratory)] were bred with female parvalbumin-cre mice (Jackson Laboratory). This mouse line is termed PVJ20 throughout the current study. A total of $n = 87$ PVJ20 mice were used in this study ($n = 44$ PVJ20 + , $n = 43$ PVJ20−). $n = 31$ mice were implanted with fiber optic implants to perform combined optogenetics and freely moving behavior, $n = 6$ mice were implanted with linear silicon probes, $n = 40$ mice were implanted with electrode arrays, and $n = 10$ mice were used for patch-clamp electrophysiology.

**Adeno-associated viral vectors**. Adeno-associated viruses (AAV) of serotype dj (hybrid capsid created from eight different AAV serotypes) were obtained from the Vector Core Facility at Oregon Health and Science University in Portland, Oregon. The AAVdj virus contained a channelrhodopsin-2 with the E123T mutation for ultrafast control (ChETA construct) fused to eYFP in a double-floxed, inverted, open-reading-frame (DIO) driven by the EF1α promoter (5.2 E + 13 vg/ml genomic titer). An eYFP construct without the ChETA sequence was used as a control (termed YFP control in this manuscript).

**Surgical procedures**. Mice were anesthetized with isoflurane (5% induction, 0.5–2% maintenance) and placed in a stereotaxic frame (Stoelting). The skull was completely cleared of all connective tissue. AAVdj-ChETA or eYFP was delivered into the medial septum (0.6 µl at 1 nL/s), at the following coordinates based on reference mouse stereotaxic atlas[69]: anteroposterior (AP) 0.85 mm from bregma, mediolateral (ML) 0 mm, dorsoventral (DV) − 4.50 mm using a 5° angle in the ML plane. Two weeks post-injection, mice were anesthetized for implantation surgery. In all, 200 µm diameter fiber optic with ceramic ferrule (Thor labs) implants were implanted at the same coordinates as for injections and cemented in place using C&B-Metabond (Patterson dental, MN). For mice used for electrophysiology experiments, an array of seven tungsten microelectrodes (~ 1 MΩ impedance) was lowered in dorsal CA1 spanning through stratum pyramidale (pyr), stratum radiatum (rad), and stratum lacunosum molecular (lm). Screws placed in the bone above the frontal cortex and cerebellum served as ground and reference, respectively. Following electrode, ground and reference placement, dental cement was applied to secure the implant permanently to the skull. Black nail polish was applied over the dental cement to block light emission during optogenetic stimulation.

**In vivo electrophysiological recordings**. Following 1 week of post-surgical recovery and one week of habituation to the tethering setup, LFP from implanted mice was recorded. All recorded signals from implanted electrodes were amplified by the tether pre-amplifier before being digitized at 22 KHz using a digital recording system (Neuralynx, USA).

**Optogenetic stimulation**. Laser stimulation was delivered through a fiber optic cord (200 µm diameter) using a laser diode fiber light source (Doric Lenses, Canada). Light intensity was calibrated and wavelength-corrected using the Power Meter Bundle with the PM100D Console and S130C Slim Photodiode Sensorlight (Thorlabs) both in vitro and in vivo. Unless stated differently, every stimulation

was performed at 50% duty cycle, which allowed the use of identical total illumination duration at all stimulation frequencies. To compute input–output (IO) curves, mice were allowed to freely explored a novel open-field and received stimulation with a 5 s ON, 5 s OFF schedule. When applying optogenetic stimulation during behavior, a loose piece of heat shrink tubing was fitted around the junction between the patch cord and the mouse' ferrule implant to limit visible light emission. Light intensities are expressed as nominal power, as measured at the tip of the fiber optic implant—cord assembly, and corrected for each wavelength.

**Novel object place recognition**. On the first day, mice were allowed to freely explore a 45 × 45 cm dark gray open-field that contained visual cues (large white horizontal and vertical gratings) on its walls for 10 min. On the second day, two identical objects (helping hand base) were presented for 10 min. On the third day, mice were allowed to explore the same open field and the location of one of the two objects was displaced. To control for potential spatial preference biases, both the initial as well as the displaced position of objects was randomized. Mice were attributed a random testing order that was kept identical throughout the three days of testing. Behavior was recorded with a video camera and analyzed offline using Optimouse, an open-source Matlab toolbox that allows for the detection of both mouse body and nose position[70]. Behavioral analysis was performed blind to the genotype and treatment. Object explorations are defined as epochs where mice have their nose within 1 cm of an object.

**Barnes maze task**. To circumvent the inability of tethered mice to perform the standard Barnes maze task (escape through a hole) we used an appetitive version. Mice were water scheduled (2 h per day) and trained to seek a 50 µL, 10% sucrose water reward in one of the 20 locations (~ 2 cm cups) of a 1 m diameter dark gray circular platform. Mice performed three trials per day for a total of five consecutive days. Spatial errors were defined as the exploration of an unbaited location (the maximum number of spatial errors is 19). On the 6th day, mice were subjected to a probe test (150 s), which consisted of a free exploration in absence of any reward. Mice exploration was recorded and the percent of time in the target quadrant was computed (chance level at 25%).

**Delayed non-match to place task**. Mice were water scheduled (2 h per day) and trained in a Y-maze to a delayed non-match to place task. In brief, each trial was divided into two phases: sample and test. In the sample phase, one arm was blocked and mice were forced to explore the opposite arm where they received a 10% sucrose water reward. In all, 30 s after completing the sample phase, mice were placed back in the central arm for the test phase, during which both arms could be explored. In the test phase, only the opposite (unexplored) arm is baited, so that mice have to alternate locations between sample and test phases. Mice were subjected to 10 trials (sample + test) per day, for 10 consecutive days, and the daily success rate was calculated at the number of correct choices divided by the total number of trials.

**Histology**. After completion of behavioral testing, mice were deeply anesthetized with a mixture ketamine/xylazine/acepromazide (100, 16, 3 mg/kg, respectively, intraperitoneal injection). For mice implanted with electrodes, microlesions were performed by passing a current of ~ 10 µA for 10–20 s through each electrode using a Nano Z (Neuralynx, USA) to identify electrode tips. Mice were then perfused transcardially with 4% paraformaldehyde in PBS (PFA). The brains were extracted and postfixed overnight in PFA at 4°C and subsequently washed in PBS for an additional 24 h at 4°C. Brains and sections were cryoprotected in solution of 30% ethylene glycol, 30% glycerol, and 40% PBS until used. Each brain was then sectioned at 50 µm using a vibratome: every section was sequentially collected in four different 1.5 mL tubes to allow different analyses (electrode location, plaque analyses, immunohistochemistry).

**Immunohistology**. Using one tube of collected brain sections (25% sampling), sections were washed 3 × 5 min in PBS to remove cryoprotective solution. Sections were first incubated overnight with PGT (0.45% Gelatin and 0.25% Triton in PBS) at 4°C. Next, slices were incubated with primary antibodies (1:1000 rabbit anti-GFP from Life Technologies, and 1:500 mouse anti-parvalbumin monoclonal IgG1 from Sigma-Aldrich) in PGT at room temperature for 2 h. Following washes of 10, 20, and 30 min, sections were then incubated with secondary antibodies [1:1000 goat anti-rabbit coupled with Alexa 488 (Molecular Probes), and 1:500 goat anti-mouse IgG1 coupled to Alexa 555 (Life Technologies)] in PGT for 45 min. Following 10, 20, and 30 min washes, sections were then mounted on glass slides and permanently coverslipped with Fluoromount mounting medium that contained 4′,6-diamidino-2-phenylindole. Only mice with histologically confirmed placement of at least one electrode in CA1 *stratum lacunosum moleculare* and optic fiber placement as well as proper construct expression in the medial septum were used in the present study. To control for co-expression with cholinergic neurons, we performed double fluorescent immunohistochemistry for eYFP and Choline acetyltransferase (ChAT). In that case, sections were incubated for 12 h with primary antibodies (1:200 goat anti-ChAT from Millipore and 1:1000 chicken anti-GFP from ThermoFisher Scientific) in PGT at 4°C. Sections were then washed in PGT and incubated with secondary antibodies (1:2000 donkey anti-goat coupled with

Alexa 647, and 1:2000 donkey anti-chicken coupled with Alexa 488 from Jackson Immunoresearch) for 45 min at room temperature. Sections were washed and mounted as described above.

**Thioflavine S staining**. To confirm the presence of oligomeric and non-oligomeric plaques in our newly generated PVJ20 mouse line, performed thioflavine S staining on brain tissue. In brief, coronal, 50 µm sections were mounted on glass slides and left to dry for ~ 1–2 h. Slides were then immersed in PBS containing 1% thioflavine S for 8 min. After clearing with absolute ethanol and PBS, slides were coverslipped and Thioflavine S positive plaques were imaged with a fluorescence microscope using a GFP dichroic cube.

**In vitro patch-clamp electrophysiology**. Mice were deeply anesthetized using ketamine/xylazine/acepromazine mix, and intracardially perfused with *N*-methyl-d-glutamine (NMDG) recovery solution (4°C) oxygenated with carbogen (5% $CO_2$ and 95% $O_2$). NMDG solution contains the following (in mM): 93 NMDG, 93 HCl, 2.5 KCl, 1.2 $NaH_2PO_4$, 30 $NaHCO_3$, 20 HEPES, 25 glucose, five sodium ascorbate, two thiourea, three sodium pyruvate, adjusted pH 7.4 with HCl before adding 10 $MgSO_4$ and 0.5 $CaCl_2$. Following NMDG perfusion, brains were quickly removed and immersed for an extra 1 min in cold NMDG recovery solution. Coronal slices (300 µm) were cut using a vibrating microtome (Leica-VT1000S), then collected in 32°C NMDG recovery solution for 10–12 min. Slices were transferred to room temperature and oxygenated artificial cerebrospinal fluid (aCSF) containing the following (in mM): 124 NaCl, 24 $NaHCO_3$, 2.5 KCl, 1.2 $NaH_2PO_4$, 2 $MgSO_4$, 5 HEPES, 2 $CaCl_2$, and 12.5 glucose (pH 7.4). Patch pipettes (3–5 MΩ) were filled with internal solution, containing the following (in mM): 140 K gluconate, 2 $MgCl_2$, 10 Hepes, 0.2 EGTA, 2 NaCl, 2 mM $Na_2$-ATP and 0.3 mM $Na_2$-GTP, adjusted pH to 7.7 with KOH, adjusted osmolarity to 290. Slices were transferred to a submerged recording chamber filled with aCSF (4 ml /min flow rate, 30 °C), continuously oxygenated with carbogen (5% $CO_2$ and 95% $O_2$). Single whole cell patch-clamp recordings of parvalbumin neurons (identified by YFP fluorescence) were done after transferring brain slices to the recording chamber. All recordings were done at 32 °C. For recordings, parvalbumin interneurons were either tested in current clamp mode (resting potential between −60 and −70 mV) or maintained at −70 mV in voltage-clamp mode. For both, the effect of different intensities of red light (638 nm) and blue light (450 nm) was tested (1–20 mW), using a laser diode fiber light source (Doric Lenses, QC, Canada). No spikes could be elicited using 450 nm stimulation in eYFP (control) transfected cells, as well as using 638 nm stimulation in eYFP-transfected cells. Electrophysiological signals were amplified, using a Multiclamp 700B patch-clamp amplifier (Axon Instruments, CA, USA), sampled at 20 kHz, filtered at 10 kHz. Data were analyzed using pClamp 10 (Molecular devices, CA, USA).

**Analysis**. Data analysis was performed using custom written Matlab codes available upon request. For silicon probe data, current source density (CSD) was calculated from 16-channel silicon probe LFP recordings using Equation (1)

$$CSD_{(x,t)} = \frac{\sigma(2x_t - (x + \Delta x_t) - (x - \Delta x_t))}{(\Delta x)^2} \qquad (1)$$

where $x_t$ is the potential at depth $x$ for time point $t$, with $\Delta x = 50$ µm. Conductivity ($\sigma$) was assumed constant. Wavelet convolution was applied to LFP signals using complex Morlet wavelets ('cmor1-1.5' in Matlab) when both time- and frequency-domain accuracy was needed, and in particular when natural gamma oscillations had to be visualized and for theta power-sorted spectrograms in Figs. 1 and 4. Moving window Fourier convolution (2 s window in the theta band, 5 s window in the gamma band, 10 ms moving steps) was used when frequency-domain accuracy was privileged over time-domain accuracy.

**Phase-amplitude coupling**. For analyses of coupled oscillations, theta phase was obtained by bandpass filtering signals between 4 and 12 Hz and performing a Hilbert transform. Although the Hilbert transform is most commonly used to obtain phase of oscillations, it was found to dampen the asymmetry that is naturally observed in theta rhythms[46]. To control for this effect, we also computed theta phase using the method described in Belluscio et al.[46]. In brief, raw traces were filtered between 1 and 60 Hz and peaks, trough, as well as zero crossing points were determined from these traces and linearly interpolated to provide phase information. In 2 s, non-overlapping sliding windows, theta phase obtained from either method was then binned in 40 bins of 9° and slow as well as fast gamma power was then averaged for each theta phase bin. Typically, most studies perform z-scoring of filtered gamma oscillation derived power to analyze coupling. We found that doing so often scales up/down phase-amplitude coupling values and mask differences in gamma power between groups. Therefore, in the current study, z-scoring is performed on the raw, unfiltered trace only in the context of phase-amplitude coupling. This method allows to reveal both increase in coupling and baseline gamma power described in Fig. 3. We also used more traditional measures such as the modulation index (MI) that was computed as the Kullback–Leibler distance between binned gamma amplitudes across theta phases and a linear distribution of gamma amplitudes[45]. To assess effects of laser stimulation in the theta band, we used entrainment fidelity[13] as a measure of how much the stimulated frequency (± 0.5 Hz) is

represented among the theta band. For instance, entrainment fidelity of 60% for a 10 Hz stimulation means that 60% of theta band is found between 9.5 and 10.5 Hz.

**Statistics**. All data are presented as mean ± standard error of the mean (SEM) and statistic test details are described in the corresponding results. All $t$ tests were two-tailed. Normality distribution of each group was assessed using Shapiro–Wilk normality test and parametric tests were used only when distributions were found normal (non-parametric tests are described where applicable). Effect size of repeated measure ANOVA was assessed with partial $\eta^2$ using Equation (2)

$$\text{partial } \eta^2 = \frac{\text{SS}_{\text{effect}}}{\text{SS}_{\text{effect}} + \text{SS}_{\text{error}}} \qquad (2)$$

1ANOVA: one-way ANOVA; 2ANOVA: two-way ANOVA; RM-1ANOVA: repeated measure one-way ANOVA. RM-2ANOVA: repeated measure two-way ANOVA. $p < 0.05$ was considered statistically significant. *, $p < 0.05$; **, $p < 0.01$; ***, $p < 0.001$, ****, $p < 0.0001$.

**Reporting summary**. Further information on research design is available in the Nature Research Reporting Summary linked to this article.

## Code availability
All the codes used in the current study are available from the corresponding author upon request.

## Data availability
The data supporting the findings of this study is available with the article and its Supplementary Information file, or is available from the corresponding author upon request. The source data underlying Figs. 1f–h, j–m, 2d–h, k–n, 3c–j, 4c–f, j–l, n, and Supplementary Figs 1c–e, 2b, d, f, 3f, 4e–g, 5b, c, e, g, 6d–h, 7d–f, 9b–d, 10f, h, j, 11b, c are provided as a Source Data file.

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

## Acknowledgements

We thank Mark Brandon and Joseph MacNeil for helpful comments on the manuscript. Ke Cui for help maintaining the colony and perfusing mice. This work is supported by Brain Canada, Fonds de la recherche en Santé du Québec (FRSQ), the Canadian Institutes of Health Research (CIHR), the Natural Sciences and Engineering Research Council of Canada (NSERC), and the Alzheimer Society of Canada. GE was supported by Brain Canada, the Roger J. Paiement Outreach Award and the Douglas Institute Foundation.

## Author contributions

G.E. and S.W. designed the study. G.E. performed and analyzed in vivo optogenetic and behavioral experiments. S.V.D.V. and I.Z. performed immunohistological experiments. I.Z. and S.V.D.V. analyzed immunohistological data. F.M. and E.T.D. performed and analyzed patch-clamp in vitro electrophysiology. G.E. and S.W. wrote the manuscript with inputs from all collaborators.

## Competing interests

The authors declare no competing interests.
