## [Peer Review File · Nature Communications]

Reviewers' comments:

Reviewer #1 (Remarks to the Author):

The authors show that optogenetic stimulation of medial septum (MS) neurons at 40 Hz can rescue retrieval performance in a novel object place recognition task in the J20 mouse model of Alzheimer's disease (AD). They first show that hippocampal gamma power is reduced in J20 mice, then show that hippocampal local field gamma oscillations can be entrained by optogenetic stimulation of MS PV cells, and finally show that gamma-frequency optogenetic activation of these cells restores to control levels the preferential exploration of a displaced object in J20 mice.

The observation that hippocampal gamma can be entrained by MS PV cells, similar to equivalent observations in the neocortex (Kim et al., PNAS 2015; not cited), is interesting, and should probably be analyzed in more detail, as described below. Importantly, MS PV cells project exclusively to gabaergic interneurons, and not principal cells, in both the hippocampus and entorhinal cortex (Unal et al., 2015), and thus have a disinhibitory effect. This may complicate the possible interpretations of the data, as outlined below.

If these caveats are properly addressed, I believe this paper would be of great interest to colleagues and make an important contribution to the field.

MAJOR POINTS:

1. Cell-type specificity

The authors used viral transduction of DIO-ChETA, and controlled the expression by using PV-cre mice. How confident are they that expression is exclusively in PV gabaergic projection neurons? Figure 2b3 seems to show that only two of the four or five eYFP-expressing cell bodies visible in the ChETA-eYFP image were PV immunopositive, and although this is just an example, it shows that this problem warrants further analysis. Their analysis in Extended figure 2 reports that transfection was not significantly different between groups, but I did not manage to extract what proportion of ChETA-eYFP cells are not PV immunopositive and this needs to be explicitly stated to enable the reader to interpret the results. If some cholinergic neurons express ChETA, one might argue that it is the cholinergic enhancement, rather than inhibitory gamma, that is responsible for the behavioural effect. This is an important consideration because cholinergic mechanisms are impaired in AD and required for object place recognition memory (Barker & Warburton, Learning & Memory 2009; Cai et al., Neurosci Lett 2012; not cited).

2. Frequency specificity

The authors attribute the behavioral effect to enhanced gamma oscillations in the hippocampus. However, there is no direct evidence for this interpretation. First, the investigators have only tested one frequency of stimulation in their behavioral task, and in order to claim frequency specificity they need to test at least one other frequency (e.g. 10 Hz as they did in Fig. 2 in tethered mice) and/or random stimulation with the same number of stimuli. Moreover, since it is known that MS inhibitory projection neurons disinhibit the hippocampus, it is difficult to exclude the possibility that the behavioural effect is due to disinhibition rather than gamma oscillations. This possibility should at least be discussed.

3. The authors use the fast channelrhodopsin mutant ChETA to enable neurons to follow 40-Hz stimulation, but do not show data that the neurons are indeed able to follow this frequency. In Extended figure 4, they use 10 Hz rather than 40 Hz to measure entrainment fidelity. They should also show to what extent the transduced PV neurons follow the optical stimulation at 40 Hz.

4. It would be useful if the authors could present CSD analysis of the gamma oscillations entrained by MS PV stimulation, so it could be compared to the physiological gamma oscillations shown in Fig. 1. This is important because there are three gamma generators converging in the CA1: one located in CA3, one in the mEC, and one in the CA1 itself (Schomburg et al., Neuron 2014; not cited). Such an analysis could give insight into which gamma generator is entrained by MS PV

neurons.

MINOR POINTS:

1. I was intrigued as to why the optogenetically induced Fourier spectrogram during behaviour (Fig. 3i; gamma peak well below 40 Hz) appears to differ from spectrograms in mice not being behaviorally tested (Fig. 2f and 3b; gamma peak at exactly 40 Hz). Could the authors offer an explanation?

3. If Fig. 3h is representative, I remain unconvinced about the difference in location preference for the animals in the test phase: In both cases the mouse appears to spend most time in the corner adjacent to the displaced object. Please explain. Also, the statistical question is not whether there is a significant discrimination index or not in each case (one-sample t-test), but whether the three groups of animals have different discrimination indices (ANOVA). [Btw., there is an error in reporting $61 \pm 0.03\%$ should be either $61 \pm 3\%$ or 0.61 ± 0.03 , and equivalent for $66 \pm 0.05\%$]

4. There are numerous typos, mis-spellings and missing words throughout the manuscript.

Reviewer #2 (Remarks to the Author):

In this manuscript, Etter et al. show that optogenetic stimulation of PV neurons at 40 Hz in medial septum drive 40 Hz oscillations in CA1 in a PVJ20+ mouse model. The authors claim that this 40 Hz optogenetic stimulation of medial septum PV neurons during the testing phase of a novel object place recognition task recues spatial memory in this PVJ20+ mouse model of AD. Overall this work seems novel and interesting, however there are significant additional experiments that need to be done to support the authors' claims. In particular more behavior and stimulation controls are required.

1. A thorough introduction of the work and its context in the prior literature is lacking. There is minimal discussion about the data reported, its implications or shortcomings. There is no discussion about the type of AD model used, or applications to other models, or information about spatial deficits previously found in this model.

a. The authors should clearly state what is novel about their approach and findings, versus what is known.

b. The authors should include a discussion of the J20 mouse model of AD.

c. The authors should provide a more balanced assessment of the literature in regard to retrieval vs. episodic memory in AD. Many papers that have assessed memory in mouse models of AD and these need to be discussed. For example, the authors state "Previous studies found that retrieval, rather than encoding of episodic memories is impaired in AD conditions" and then they cite only one paper.

2. Key behavior controls should be performed in order to support the claim that spatial recognition memory is being rescued by 40 Hz optogenetic stimulation.

a. In Figure 3k-l only 1 of 2 behavior metrics analyzed during the test phase show a deficit between the J20+ and J20- YFP groups. Further behavior tasks with deficits between J20+/- groups are necessary to show if this rescue is robust.

b. The author should assess key behavior controls like activity level, anxiety, motivation etc. to determine if the improvement in spatial recognition is due to some non-specific effects.

3. More stimulation controls should be performed to control for the non-specific effects of optogenetics and to determine if the effects are frequency specific.

a. Other frequencies of stimulation, such as 8 Hz, 20 Hz, and 80 Hz, should be assessed. Additionally, a random stimulation control is important to control for the overall amount of stimulation.

- b. What is the duty cycle of the optogenetic pulses during different frequencies? (10 Hz, 40 Hz, 8x40 Hz burst) Optogenetic stimulation has typically been short (like 1ms) to reduce potential side effects like heating and calcium influx. If longer, then why?
- c. The methods section should include a thorough description of the optogenetics.

4. The analysis of the electrophysiological data is somewhat lacking.

- a. The authors should acknowledge that the gamma oscillations to which they are referring, and driving, are slow gamma, as it is now widely acknowledged that there are multiple gammas in the hippocampus. They could state once at the beginning that this is slow gamma and state they will be calling it "gamma" from there on out.
- b. More information should be provided on how the LFP traces were Z-scored. Over what time periods and what behavior states? There could be very different effects between animals if some are mostly running (theta), versus mostly still (non-theta).
- c. Axes and color bars need labels and units throughout. Simply putting min/max for color bars is confusing and does not give the reader any idea of the scale of these effects.
- d. Figure 1k is confusing without labels on the y-axis (the y-axis could be frequency instead of distribution across hippocampal layers).
- e. Co-modulograms, phase vs. frequency, in PVJ20+/- mice should also be shown in Figure 1
- f. Figure 1 m,n,p,q: shading should be somewhat translucent so that the reader can visualize the overlap between distributions – this is important for understanding differences and similarities between distributions .

5. The optogenetic stimulation is interesting but more characterization of this stimulation in vivo is needed.

- a. I believe the electrophysiology data shown in Figure 2 is from in vivo experiments but it is not stated explicitly. Please state explicitly when experiments are in vivo vs in slice.
- b. The authors should show PSD across all frequencies (2-60 Hz) during baseline, 10 Hz, 40 Hz, and 8x40 Hz stimulation.
- c. Furthermore, the authors should stimulate with the same duration as during the behavior experiments (10 min of 40 Hz stimulation) and show the spectrograms during that stimulation. Note that prolonged stimulation can have different effects than short stimulation, for example more calcium will flood into the cell with longer stimulation.

6. It is especially surprising that stimulation of PV interneurons in septum does not affect theta oscillations in the hippocampus considering the role of septum in hippocampal theta. This needs to be examined more thoroughly.

- a. Is this because LFP is z-scored? What happens in traces that are not z-scored?
- b. Are other aspects of theta affected, like phase, frequency, or theta phase modulation of spiking?
- c. How do the authors explain this result considering what is known about the septum?

Reviewer #3 (Remarks to the Author):

The study by Etter and colleagues showed that reinstating slow gamma oscillations within hippocampal networks in a modified mouse model of AD restore memory deficits in an object-location task.

This is a really interesting story with innovative ideas and I really liked the concept. However, there is substantial concerns that dampened my enthusiasm.

Before entering into specific comments, I think that the article will highly benefit a re-writing in a lengthier way, as a lot of shortcuts are present in the manuscript and some strong claims are not always backed-up by actual data.

I have some major comments, mainly related to the analysis and interpretation of the data:

Major comments:

1- The authors characterize gamma oscillation in the CA1 area of the dorsal hippocampus in the PVJ20+ vs PVJ20- mice and showed a decrease coupling between theta phase and what they called gamma oscillations (30-60 Hz). They reported "Strongest gamma (30 – 60 Hz) CSD power was found between the pyramidal cell and the lacunosum moleculare (Im) layers of the hippocampal CA1 field of PVJ20- mice (fig. 1i-j). Furthermore, gamma oscillations were most strongly coupled to theta rhythms in the Im layer (fig. 1k). We thus focused the following analyses in larger groups of mice in this layer."

I clearly have a problem with these results:

- a. First of all, these data were analyzed from 1 animal (as stated in the methods, just 1 animal of each group -recorded with silicon probe). How the statistics were done then?
- b. It is well known that slow gamma (which is what the authors looked at) is strongest in the CA1 radiatum, as it is coming from CA3 (I will not list all the studies confirming this result which includes papers from the Buzsaki's group, the Dupret' group, the Moser's group and much more...). How the authors can find, using CSD analysis, a higher coupling in the LM, where medium (or fast, depending on the authors) gamma is the strongest and not slow gamma?
- c. In the figure 1l, and as expected from literature cited above, it is clear from the wavelet convolution that the LM mainly exhibit medium gamma (centered around 60 Hz in PVJ20-). As such, one can see that medium gamma is decreased in PVJ20+ animals compared to PVJ20-. What is the spectral content of the recordings in the LM in both groups (a simple power spectrum would be nice).

2- The authors stated that restoring slow gamma oscillations during the retrieval phase is sufficient to rescue memory deficits. However, (and except misreading from my side), they never actually showed that slow gamma oscillations are indeed altered during the retrieval phase in the PVJ20+ animals. This is an important piece of data that is missing in the present study.

3- What is the effect of PV stimulation on medium gamma power and coupling to theta phase in the LM? As stated above, medium gamma is mainly present in the LM and coming from the EC. Is this gamma band also increased following septal stimulation?

More globally, what is the effect of such stimulation in the different hippocampal layers covered by the tungsten micro-electrodes

4- The authors stated that "optogenetic stimulation of parvalbumin neurons at 40 Hz - but not other frequencies restore gamma oscillations in hippocampus of the J20 AD mouse model". However, i) they only test for slow gamma (30-60Hz) in hippocampal recordings and ii) they only test 40 Hz stimulations (in addition to theta burst and 10 Hz stimulation). As such, the authors only showed that a 40Hz stimulation of septal PV cells induced a 40Hz gamma rhythm in the LM. What about other gamma frequencies (30, 50, 60, and 80 Hz for example?)

5- The authors used the Hilbert transform of the filtered theta trace to determine theta phase. However, it is well known that Hilbert transform will not give the "true" phase of the oscillation, as theta is not sinusoidal (see, for example Belluscio study). Is it possible that part of the results was indeed due to a change in theta asymmetry? Related to that, the presence of strong gamma rhythm will bias the phase calculation. I recommend the authors to use the Belluscio's method to dampen this effect.

Minors points:

1- The authors used a "new" line of AD mice: the PVJ20 mice. They describe the presence of amyloid plaque and deficits in the object location in 6 months old animals. However, the authors did not test whether these mice were similar to the APPJ20 mice (at behavioral and molecular levels). A more complete description of the new mouse line is needed to fully validate the results.

2- All the data are expressed in figure in a "min-max" scale. This can be really misleading, as no "real" numbers are reported.

3- I highly recommend the authors to extend the article. Too many shortcuts are present in the manuscript.

4- I found odd that the authors do not cite one time the article from Martinez-Losa and colleagues (2018) who showed, in the same mouse line, that restoration of cortical gamma oscillations rescued memory deficits.

Reviewers' comments:

Reviewer #1 (Remarks to the Author):

The authors show that optogenetic stimulation of medial septum (MS) neurons at 40 Hz can rescue retrieval performance in a novel object place recognition task in the J20 mouse model of Alzheimer's disease (AD). They first show that hippocampal gamma power is reduced in J20 mice, then show that hippocampal local field gamma oscillations can be entrained by optogenetic stimulation of MS PV cells, and finally show that gamma-frequency optogenetic activation of these cells restores to control levels the preferential exploration of a displaced object in J20 mice.

The observation that hippocampal gamma can be entrained by MS PV cells, similar to equivalent observations in the neocortex (Kim et al., PNAS 2015; not cited), is interesting, and should probably be analyzed in more detail, as described below. Importantly, MS PV cells project exclusively to gabaergic interneurons, and not principal cells, in both the hippocampus and entorhinal cortex (Unal et al., 2015), and thus have a disinhibitory effect. This may complicate the possible interpretations of the data, as outlined below.

If these caveats are properly addressed, I believe this paper would be of great interest to colleagues and make an important contribution to the field.

We thank reviewer 1 for the helpful comments. We are now citing Kim et al., 2015 which is indeed relevant to our study. We also added analysis of effects of MSPV cells in more details as suggested by the reviewer (see below)

MAJOR POINTS:

1. Cell-type specificity

The authors used viral transduction of DIO-ChETA, and controlled the expression by using PV-cre mice. How confident are they that expression is exclusively in PV gabaergic projection neurons? Figure 2b3 seems to show that only two of the four or five eYFP-expressing cell bodies visible in the ChETA-eYFP image were PV immunopositive, and although this is just an example, it shows that this problem warrants further analysis. Their analysis in Extended figure 2 reports that transfection was not significantly different between groups, but I did not manage to extract what proportion of ChETA-eYFP cells are not PV immunopositive and this needs to be explicitly stated to enable the reader to interpret the results. If some cholinergic neurons express ChETA, one might argue that it is the cholinergic enhancement, rather than inhibitory gamma, that is responsible for the behavioural effect. This is an important consideration because cholinergic mechanisms are impaired in AD and required for object place recognition memory (Barker & Warburton, Learning & Memory 2009; Cai et al., Neurosci Lett 2012; not cited).

This is indeed an important point. Non-specific cholinergic activation by ChETA could potentially account for the behavioral effects observed in our study. However, we have performed an extensive histological analysis of cell specificity of our transfections and our results show that non-specific transfection in cholinergic neurons was absent (supplementary fig. 4). We quantified the expression of ChETA in cholinergic cells by performing a Choline Acetyltransferase (ChAT) immunohistochemistry in PVJ20+/PVJ20- mice sections injected

with ChETA-eYFP or the eYFP control construct. Our results show that there was virtually no expression of either construct in cholinergic cells (see supplementary fig. 4d-g) suggesting that the behavioral effects were not due to cholinergic cell activation.

We also acknowledge that representing data in terms of (% total counted cells) is not very informative. We now display % of PV/ChAT cells transfected as well as % of eYFP cells that are either PV or ChAT (supplementary fig. 4e-g). As for the apparent discrepancy in the count between PV-and ChETA-YFP+ cells, it is very likely that the localization of PV and ChETA-YFP is not on the same plane as PV expression is cytoplasmic while ChETA-YFP is expressed only on the membrane.

2. Frequency specificity

The authors attribute the behavioral effect to enhanced gamma oscillations in the hippocampus. However, there is no direct evidence for this interpretation. First, the investigators have only tested one frequency of stimulation in their behavioral task, and in order to claim frequency specificity they need to test at least one other frequency (e.g. 10 Hz as they did in Fig. 2 in tethered mice) and/or random stimulation with the same number of stimuli. Moreover, since it is known that MS inhibitory projection neurons disinhibit the hippocampus, it is difficult to exclude the possibility that the behavioural effect is due to disinhibition rather than gamma oscillations. This possibility should at least be discussed.

We agree with the reviewer that it is essential to assess frequency specificity. To this end, we now tested mice stimulated with 80 Hz (to test for frequency specificity), but also 40 Hz during the sample phase (to test for memory retrieval specificity; fig 4). We found that we could only significantly improve memory with 40 Hz stimulation during the retrieval phase, while 80 Hz stimulation during retrieval or 40 Hz stimulation during encoding (sample phase)

did not significantly ameliorate memory (fig 4n). Secondly, from a theoretical standpoint, we may indeed release feedforward inhibition onto hippocampal principal cells when stimulating MSPV cells. However, the study from Zutshi et al., 2018 (cited) showed reduced hippocampal place cell firing frequency during MSPV pacing. Moreover, a simple disinhibition is unlikely to be the key effect since the 40 Hz stimulation increased only power at 40 Hz as well as its coupling to theta phase whereas either the theta or high gamma power were not affected (fig. 2k-m; supplementary fig. 13d,e). In contrast, a simple disinhibition should have affected all frequencies equally. Moreover, a further analysis of the effect of the 40 Hz opto stim on interneuron and pyramidal cell spiking could provide further clarification as to all the effects of the 40 Hz stims, and we have indicated these limitations in the discussion as suggested by the reviewer (line 235).

3. The authors use the fast channelrhodopsin mutant ChETA to enable neurons to follow 40-Hz stimulation, but do not show data that the neurons are indeed able to follow this frequency. In Extended figure 4, they use 10 Hz rather than 40 Hz to measure entrainment fidelity. They should also show to what extent the transduced PV neurons follow the optical stimulation at 40 Hz.

We agree that this is an important point. The advantages of 10 Hz stimulation is that it is within the theta band - as such, the 'entrainment fidelity' (Bender et al., 2015 - cited) describes how well we can pace theta at that exact frequency, which is harder to compute with frequencies that are outside of the theta band, including 40 and 80Hz since gamma PSD tend to be much smaller when using Fourier transforms. To directly answer this question, we now show (1) results obtained from whole cell recordings of MSPV cells transfected with ChETA being paced at 40 and 80 Hz *in vitro* (fig. 2a-h), and (2) the corresponding spectrograms in

vivo (fig. 2j). Finally, we also computed input/output (IO) curves for 40 Hz and 80 Hz stimulation by measuring the PSD at these frequencies and showing the corresponding entrainment (supplementary fig. 6f-h)

4. It would be useful if the authors could present CSD analysis of the gamma oscillations entrained by MS PV stimulation, so it could be compared to the physiological gamma oscillations shown in Fig. 1. This is important because there are three gamma generators converging in the CA1: one located in CA3, one in the mEC, and one in the CA1 itself (Schomburg et al., Neuron 2014; not cited). Such an analysis could give insight into which gamma generator is entrained by MS PV neurons.

We have added a supplementary figure showing recordings from a PVJ20- mouse implanted with a linear probe in CA1 and repeated optogenetic stimulation of MSPV cells with various frequencies, including 10 Hz, 40 Hz and 80 Hz (supplementary fig. 10). We now cite the article by Schomburg et al. 2014. In CSD signals, the strongest theta and gamma oscillations are found closer to the hippocampal fissure (and thus in the stratum *lm*), while Schomburg et al. performed ICA decomposition to approximate the location of gamma generators. While this question is interesting, most of our study's statistical power relies on single electrode data which does not allow ICA decomposition. We now added this limitation in our discussion (line 252).

MINOR POINTS:

1. I was intrigued as to why the optogenetically induced Fourier spectrogram during behaviour (Fig. 3i; gamma peak well below 40 Hz) appears to differ from spectrograms in

mice not being behaviorally tested (Fig. 2f and 3b; gamma peak at exactly 40 Hz). Could the authors offer an explanation?

There was indeed a scale issue that has now been fixed, and ticks have also been added, and these results are now presented in fig 4h. We apologize for the lack of clarity in the previous version of the manuscript.

3. If Fig. 3h is representative, I remain unconvinced about the difference in location preference for the animals in the test phase: In both cases the mouse appears to spend most time in the corner adjacent to the displaced object. Please explain. Also, the statistical question is not whether there is a significant discrimination index or not in each case (one-sample t-test), but whether the three groups of animals have different discrimination indices (ANOVA). [Btw., there is an error in reporting $61 \pm 0.03\%$ should be either $61 \pm 3\%$ or 0.61 ± 0.03 , and equivalent for $66 \pm 0.05\%$]

We agree that in the previous version, these occupancy plots were not very informative about the behavior of the mouse. We have now plotted the location of the mouse trajectory across the maze to show its exploration (which usually covers the entire maze), and overlaid a heatmap of object exploration (thus excluding nose position away from the objects). This data is now presented in fig 4m. Errors in reporting have also been fixed.

4. There are numerous typos, mis-spellings and missing words throughout the manuscript.

We thank the reviewer and have corrected these typos

Reviewer #2 (Remarks to the Author):

In this manuscript, Etter et al. show that optogenetic stimulation of PV neurons at 40 Hz in medial septum drive 40 Hz oscillations in CA1 in a PVJ20+ mouse model. The authors claim that this 40 Hz optogenetic stimulation of medial septum PV neurons during the testing phase of a novel object place recognition task recues spatial memory in this PVJ20+ mouse model of AD. Overall this work seems novel and interesting, however there are significant additional experiments that need to be done to support the authors' claims. In particular more behavior and stimulation controls are required.

We thank reviewer #2 for underlining the novelty of our study and for the specific comments and suggestions.

1. A thorough introduction of the work and its context in the prior literature is lacking. There is minimal discussion about the data reported, its implications or shortcomings. There is no discussion about the type of AD model used, or applications to other models, or information about spatial deficits previously found in this model.

a. The authors should clearly state what is novel about their approach and findings, versus what is known.

We agree that the introduction and discussion were short as the first version of the manuscript was written for a short communication format. However in this longer version, we have now extended the introduction and highlighted the originality of the findings (line 39, line 74).

b. The authors should include a discussion of the J20 mouse model of AD.

We have now added a discussion of the AD mouse model used and compared our results to what has been found in other models (line 65, line 225)

c. *The authors should provide a more balanced assessment of the literature in regard to retrieval vs. episodic memory in AD. Many papers that have assessed memory in mouse models of AD and these need to be discussed. For example, the authors state “Previous studies found that retrieval, rather than encoding of episodic memories is impaired in AD conditions” and then they cite only one paper.*

We agree with reviewer #2, and now provide a more balanced assessment of the literature (lines 187-190)

2. *Key behavior controls should be performed in order to support the claim that spatial recognition memory is being rescued by 40 Hz optogenetic stimulation.*

We have now added an additional stimulation group at 80 Hz (fast gamma). Results can be found in figure 4d, in the result section (lines 142-162), and in the discussion (line 242).

a. *In Figure 3k-l only 1 of 2 behavior metrics analyzed during the test phase show a deficit between the J20+ and J20- YFP groups. Further behavior tasks with deficits between J20+/- groups are necessary to show if this rescue is robust.*

In the previous versions of the manuscript, we quantified the discrimination index during the sample phase as a control to show that mice do not display spontaneous preferences before the test phase. We also showed total exploration times across groups to make sure that there are no significant differences that could account for differences in discrimination. We now show the discrimination index in the sample and test phases, and used a 2 way anova to measure differences in discrimination index (fig. 4n).

We agree that more behavioral data could reinforce the findings. While we did not find a task that PVJ20 could learn that also discriminated the encoding/retrieval phases (see below), we

have instead added data on PVJ20 mice tested in a spatial reference memory task (Barnes maze, dry version) and found cognitive defects when mice were 6 months of age (supplementary fig. 2a-d; lines 90-91 of the result section). However this task does not discriminate between the encoding and retrieval phases. We then aimed to train PVJ20 mice in a delayed alternation task (that allows separate analysis of encoding vs retrieval phases) however PVJ20 mice were never able to learn the task with a minimum 30s delay, making it impossible to further assess the effects of stimulation on working memory (supplementary fig 2e,f and lines 91-92 of the results section). We believe that these behavioral data are interesting in the context of phenotypic data from the PVJ20 mouse line, which has been mentioned by another reviewer. Finally, the two new groups tested in the novel object place recognition task (80 Hz during retrieval, 40 Hz during encoding) further confirmed the specificity of 40 Hz opto stim rescue of memory during retrieval.

b. The author should assess key behavior controls like activity level, anxiety, motivation etc. to determine if the improvement in spatial recognition is due to some non-specific effects.

We agree with this suggestion. We have therefore analysed locomotor speed/distance traveled as a measure of activity, as well as thigmotaxis (% time spent in the center of an open field) as a measure of anxiety (the lower the more anxiety; supplementary fig. 11a-b). Importantly, we found that the total exploration time (displaced + familiar objects) during the test phase did not differ, suggesting that group differences cannot be explained by changes in activity levels, anxiety, or motivation (supplementary fig. 11c). These results are now discussed (lines 244)

3. More stimulation controls should be performed to control for the non-specific effects of optogenetics and to determine if the effects are frequency specific.

a. *Other frequencies of stimulation, such as 8 Hz, 20 Hz, and 80 Hz, should be assessed.*

Additionally, a random stimulation control is important to control for the overall amount of stimulation.

We now include a new 80 Hz stimulation group which allows to test the specificity of 40 Hz stimulation. Unfortunately, because of time limitations we could not generate enough mice to try additional frequencies in the context of behavioral testing (n = 5 PVJ20+ minimum per group for 3 groups, considering PVJ20+ represent half of the litters; we would need a minimum of 30-50 PVJ20 pups), but decided to test a frequency that was functionally most meaningful (80 Hz is within the fast gamma frequency band). As we are using 50% duty-cycle stimulation, the amount of light delivered is comparable between 10Hz/40Hz/80Hz stimulation. As a generalization that we are able to pace oscillations at different frequencies, we added a supplementary figure showing effects of 8 Hz, 20 Hz as well as other frequencies (5, 20, 30,50 Hz) on CA1 LFP signals (supplementary fig. 8).

b. *What is the duty cycle of the optogenetic pulses during different frequencies? (10 Hz, 40 Hz, 8x40 Hz burst) Optogenetic stimulation has typically been short (like 1ms) to reduce potential side effects like heating and calcium influx. If longer, then why?*

We used 50% duty cycle throughout the study (now added to Material and Methods, lines 4753-486) which means 12 ms for 40 hz, and 6ms for 80 Hz. We generally obtained better entrainment with these parameters and also ensured that the total amount of light integrated remained consistent between stimulation parameters (so effects cannot be accounted for by the illumination time from one stimulation pattern to another). We agree that stimulation using the original ChR2 construct can generally be shorter: 5, 10, 15 ms were tested in the princeps paper from Karl Deisseroth, 15ms being associated to better fidelity (Boyden et al., 2005 - not

cited). However, while ChETA stimulation is associated with shorter depolarization that allow for faster stimulation frequencies, ChETA was also reported to require longer stimulation pulses to elicit action potentials and is still outperformed by wild-type ChR2 at 1ms (Gunaydin et al., 2010 - now cited). We now show in vitro data for photocurrents and fidelity (fig. 2e-h) and discuss this point (lines 131-132, 239-242)

c. *The methods section should include a thorough description of the optogenetics.*

We have now added a new method section dedicated to optogenetics (lines 475-486).

4. *The analysis of the electrophysiological data is somewhat lacking.*

a. *The authors should acknowledge that the gamma oscillations to which they are referring, and driving, are slow gamma, as it is now widely acknowledged that there are multiple gammas in the hippocampus. They could state once at the beginning that this is slow gamma and state they will be calling it "gamma" from there on out.*

We agree with the reviewer that this distinction should be made in the light of recent studies.

We have now discriminated slow (30-60 Hz, and fast (60 - 80Hz) gamma in figure 1 (panels g,h,p,q,r,s), figure 2 (l,m), and figure 3 (d-i), as well as throughout the text, every time gamma oscillations were mentioned in the previous version.

b. *More information should be provided on how the LFP traces were Z-scored. Over what time periods and what behavior states? There could be very different effects between animals if some are mostly running (theta), versus mostly still (non-theta).*

This is indeed an important point for the interpretation of results and we now added a clear description in the methods section. We agree that amount of running periods (and thus theta

states) can differ from mouse to mouse. In figure 1, since we normalize gamma oscillations by z-scored theta, we needed to make sure that the distribution of theta power was comparable between animals and groups. This is what we did in figure 1k-n. We now describe precisely how traces were z-scored in the methods section (lines 617-625).

c. *Axes and color bars need labels and units throughout. Simply putting min/max for color bars is confusing and does not give the reader any idea of the scale of these effects.*

We apologize for this and have now replaced the min/max nomenclature by raw values in all the figures concerned: figure 1h,j,o; figure 2j, figure 3b,c, figure 4h as well as in all new supplementary figures..

d. *Figure 1k is confusing without labels on the y-axis (the y-axis could be frequency instead of distribution across hippocampal layers).*

We have now added depth labels on y-axes of figures concerning silicon probes (fig. 1e-h).

We have decided to replace figure 1k by comodulograms, as suggested in the next comment

e. *Co-modulograms, phase vs. frequency, in PVJ20+/- mice should also be shown in Figure 1*

We agree that comodulograms can aid at understanding which phase/frequencies are coupled so we added phase/frequency comodulograms summarizing each depth in fig. 1h. More classic comodulograms for each recording site are shown on supplementary fig. 3.

f. *Figure 1 m,n,p,q: shading should be somewhat translucent so that the reader can visualize the overlap between distributions – this is important for understanding differences and similarities between distributions .*

We agree with reviewer #2 and have now made translucent every graph area that presents some overlap.

5. *The optogenetic stimulation is interesting but more characterization of this stimulation in vivo is needed.*

a. *I believe the electrophysiology data shown in Figure 2 is from in vivo experiments but it is not stated explicitly. Please state explicitly when experiments are in vivo vs in slice.*

We apologize for this. To prevent any further confusion we are now clearly stating what the recording conditions are in a schema associated with each result (fig. 1a,b,c,i; fig. 2a,i; fig. 3a; fig 4a,g; and in all new supplementary figures).

b. *The authors should show PSD across all frequencies (2-60 Hz) during baseline, 10 Hz, 40 Hz, and 8x40 Hz stimulation.*

We agree that more detailed representation can be more convincing. We now show PSD for each recording epoch during baseline as well as 5, 8, 10, 20, 30,40, 50, 80 Hz stimulation (supplementary fig. 8)

c. *Furthermore, the authors should stimulate with the same duration as during the behavior experiments (10 min of 40 Hz stimulation) and show the spectrograms during that stimulation. Note that prolonged stimulation can have different effects than short stimulation, for example more calcium will flood into the cell with longer stimulation.*

This is indeed a crucial control. To answer this question, we have recorded 4 mice implanted with single electrodes and performed 40 Hz stimulation for 10 min as they were performing the test phase of novel place object recognition task. We did not find significant decrease in

entrainment during this length of time (fig. h-j). While this might be out of the scope of this study, we have also been able to pace Ca1-lfp in the same conditions (same construct/mouse line) for 24h periods (data not shown).

6. *It is especially surprising that stimulation of PV interneurons in septum does not affect theta oscillations in the hippocampus considering the role of septum in hippocampal theta.*

This needs to be examined more thoroughly.

We agree with reviewer #2 that this is an interesting piece of data. Importantly, we show that while 40 Hz stimulation does not decrease theta oscillation power (fig. 2k), frequency (fig. 4k), phase distribution (fig 4.l), or waveform asymmetry (supplementary fig. 9), stimulation in the theta range (eg 10 Hz stimulation) do decrease endogenous theta (that is usually at around ~8Hz). This is revealed by increased entrainment fidelity in supplementary fig. 6 (if natural theta oscillations were preserved in addition to oscillations being driven at 10 Hz, this index should stay the same during stimulation)

a. *Is this because LFP is z-scored? What happens in traces that are not z-scored?*

In addition to z-scoring gamma power as done in most studies (fig 3i), we also analyzed data where z-scoring was performed only on the LFP trace (fig 3c-g), as well as raw data that was not z-scored (supplementary fig 6). This aspect has also been described in more details in the methods section (lines 617-625).

b. *Are other aspects of theta affected, like phase, frequency, or theta phase modulation of spiking?*

While we did not look at unit activity in our study, we analyzed theta frequency and phase distribution during prolonged 40 Hz stimulation and found that neither parameter was altered (fig. 4k & l respectively)

c. *How do the authors explain this result considering what is known about the septum?*

This is indeed a great point of discussion. We have now added a new paragraph in the discussion (line 235-251)

Reviewer #3 (Remarks to the Author):

The study by Etter and colleagues showed that reinstating slow gamma oscillations within hippocampal networks in a modified mouse model of AD restore memory deficits in an object-location task.

This is a really interesting story with innovative ideas and I really liked the concept. However, there is substantial concerns that dampened my enthusiasm.

Before entering into specific comments, I think that the article will highly benefit a re-writing in a lengthier way, as a lot of shortcuts are present in the manuscript and some strong claims are not always backed-up by actual data.

I have some major comments, mainly related to the analysis and interpretation of the data:

We thank reviewer #3 for reviewing this manuscript and the enthusiasm for the first version.

We agree with this comment and formatted the manuscript to a lengthier version.

Major comments:

1- *The authors characterize gamma oscillation in the CA1 area of the dorsal hippocampus in the PVJ20+ vs PVJ20- mice and showed a decrease coupling between theta phase and what they called gamma oscillations (30-60 Hz). They reported “Strongest gamma (30 – 60 Hz) CSD power was found between the pyramidal cell and the lacunosum moleculare (lm) layers of the hippocampal CA1 field of PVJ20- mice (fig. 1i-j). Furthermore, gamma oscillations were most strongly coupled to theta rhythms in the lm layer (fig. 1k). We thus focused the following analyses in larger groups of mice in this layer.”*

I clearly have a problem with these results:

a. *First of all, these data were analyzed from 1 animal (as stated in the methods, just 1 animal of each group -recorded with silicon probe). How the statistics were done then?*

We apologize for the confusion. We have now added more mice with silicon probes to compute statistics (n = 3 PVJ20+ mice and n = 3 PVJ20- mice; fig. 1a-h). To gain statistical power, we refined our analyses in the stratum lm using single electrodes (n = 10 PVJ20+ mice and n = 10 PVJ20- mice; fig. 1i-s). We have made sure to avoid confusion by inserting panels describing the recording conditions and detailing the group sizes.

b. *It is well known that slow gamma (which is what the authors looked at) is strongest in the CA1 radiatum, as it is coming from CA3 (I will not list all the studies confirming this result which includes papers from the Buzsaki's group, the Dupret' group, the Moser's group and much more...). How the authors can find, using CSD analysis, a higher coupling in the LM, where medium (or fast, depending on the authors) gamma is the strongest and not slow gamma?*

We agree with reviewer #3, and now present a complete description of both slow and fast gamma oscillations throughout the manuscript. We now show comodulograms

(supplementary fig 3) and effects of stimulation (supplementary fig 10) across the different layers of CA1 using CSD signals. It is important to mention that to estimate the source of specific frequency oscillations, independent component analysis (ICA - used by Schomburg et al., 2014, cited, from the Buzsaki group) or other decomposition methods have to be used. While we focused most of our work on single electrodes (mainly to gain statistical power when comparing groups), this recording configuration does not give insight on the exact location of different gamma oscillations. We now discuss this limitation in the discussion (line 252).

c. In the figure 11, and as expected from literature cited above, it is clear from the wavelet convolution that the LM mainly exhibit medium gamma (centered around 60 Hz in PVJ20-). As such, one can see that medium gamma is decreased in PVJ20+ animals compared to PVJ20-. What is the spectral content of the recordings in the LM in both groups (a simple power spectrum would be nice).

We now discriminated between slow (30 - 60 Hz) and fast (60 - 120Hz) and found indeed that both types of gamma were decreased in PVJ20+ mice (fig. 1p,q) while only slow gamma phase amplitude coupling was reduced in PVJ20+ mice (fig. 1r,s). While we found that fourrier spectrograms do not have the temporal resolution to isolate single theta cycles and thus have the tendency to average periods with and without gamma oscillations (fig. 1j, left panel shows gamma 'bursts' associated with each theta cycle), we now show power spectra for each recording epochs (supplementary fig. 10; were the traditional decay of spectral power with increasing frequencies using a fourrier transform can be seen). Additionally, we also show comodulograms in every layer of CA1 (oriens, pyramidale, radiatum lacunosum moleculare) in supplementary fig. 3.

2- *The authors stated that restoring slow gamma oscillations during the retrieval phase is sufficient to rescue memory deficits. However, (and except misreading from my side), they never actually showed that slow gamma oscillations are indeed altered during the retrieval phase in the PVJ20+ animals. This is an important piece of data that is missing in the present study.*

We agree that this is a key point. We have implanted a subset of mice (n = 4 PVJ20+, n = 4 PVJ20-) with single electrode and perform the novel object place recognition task and could recapitulate the phenotype presented in fig. 1p-s. These new results are now presented in fig. 4b-f and in the results section (line 193-199)

3- *What is the effect of PV stimulation on medium gamma power and coupling to theta phase in the LM? As stated above, medium gamma is mainly present in the LM and coming from the EC. Is this gamma band also increased following septal stimulation?*

More globally, what is the effect of such stimulation in the different hippocampal layers covered by the tungsten micro-electrodes

To answer the first point, while we could drive 80 Hz stimulation in CA1 (fig 2i,m), we were not able to robustly increase the whole fast/medium gamma band power (fig 2l), nor its coupling to theta oscillations (fig 3 g) which was not found to be decreased in PVJ20+ mice (fig 1s). To answer the second point, responses to stimulation were frequency specific (i.e. we could not increase slow/low gamma with stimulation frequency outside of that frequency band). This can be observed in more details in supplementary figure 8. To answer the final point, we now show the distribution of responses with silicon probes (which span the transversal axis of CA1 more robustly than tungsten arrays) in supplementary fig. 10. As

stated above, when not using ICA decomposition, CSD power of theta and gamma oscillations is strongest in the Im, and 40 or 80 Hz stimulation induced increase in power can be observed in all layers.

4- *The authors stated that “optogenetic stimulation of parvalbumin neurons at 40 Hz - but not other frequencies restore gamma oscillations in hippocampus of the J20 AD mouse model”. However, i) they only test for slow gamma (30-60Hz) in hippocampal recordings and ii) they only test 40 Hz stimulations (in addition to theta burst and 10 Hz stimulation). As such, the authors only showed that a 40Hz stimulation of septal PV cells induced a 40Hz gamma rhythm in the LM. What about other gamma frequencies (30, 50, 60, and 80 Hz for example?)*

This is indeed an important point. We now show effects in both slow and fast gamma bands. Then, we have added a new stimulation group at 80 Hz (n = 5) and tested the effect of these stimulation in the novel place object recognition task (fig 4a,n) and found no memory improvements in the test phase with such stimulation. Finally, we now show the effect of additional frequencies on CA1 PSDs in supplementary fig. 8.

5- *The authors used the Hilbert transform of the filtered theta trace to determine theta phase. However, it is well known that Hilbert transform will not give the “true” phase of the oscillation, as theta is not sinusoidal (see, for example Belluscio study). Is it possible that part of the results was indeed due to a change in theta asymmetry? Related to that, the presence of strong gamma rhythm will bias the phase calculation. I recommend the authors to use the Belluscio’s method to dampen this effect.*

This is also an interesting point: an elegant method to detect the phase of theta was described in Belluscio et al., 2012 (now cited). We agree with the reviewer that improper phase detection could have consequences on the results. We thus employed the method described in Belluscio et al., 2012 (briefly described in the method section and in supplementary fig. 9a), measured phase amplitude coupling before and during 40 Hz optogenetic gamma stimulation and found no significant difference between this method and using the Hilbert transform (supplementary fig. 9d). While this new method is an important alternative to more classic Hilbert transform, we find that presenting both method could be of interest of the reader in order to compare for studies having used one method or the other.

Minors points:

1- *The authors used a “new” line of AD mice: the PVJ20 mice. They describe the presence of amyloid plaque and deficits in the object location in 6 months old animals. However, the authors did not test whether these mice were similar to the APPJ20 mice (at behavioral and molecular levels). A more complete description of the new mouse line is needed to fully validate the results.*

We agree that this is an important point. We have now added more data concerning the spatial memory phenotype of PVJ20 mice in both a spatial reference memory task (dry version of the Barnes maze; supplementary fig. 2a-d) and a spatial working memory task (delayed non-match to place in the Y-maze; supplementary fig. 2e-f), which is in line with the phenotype that was described in the literature (Webster et al., 2014 - not cited).

2- *All the data are expressed in figure in a “min-max” scale. This can be really misleading, as no “real“ numbers are reported.*

We agree that these labels can be misleading and have replaced min-max to actual values.

3- *I highly recommend the authors to extend the article. Too many shortcuts are present in the manuscript.*

We have now significantly extended the introduction and discussion of the manuscript.

4- *I found odd that the authors do not cite one time the article from Martinez-Losa and colleagues (2018) who showed, in the same mouse line, that restoration of cortical gamma oscillations rescued memory deficits.*

This study is indeed perfectly relevant for this manuscript, we apologize for missing this article and thank the author for pointing it out.

Reviewers' comments:

Reviewer #1 (Remarks to the Author):

The revised manuscript is in large parts rewritten and much improved.

However, I still have some concerns about the presentation. Most importantly, the authors need to tone down several of their conclusions, which are currently not supported by the data.

1. The authors suggest that 'slow gamma oscillations are essential for memory retrieval' (lines 23-24) and claim that their study 'establishes a causal link between hippocampal slow gamma oscillations and spatial memory' (lines 219-220). Neither of these statements is correct. The authors show that hippocampal slow gamma, as recorded in the stratum lacunosum-moleculare, has lower power in the J20 mouse model of Alzheimer's disease and that optogenetic stimulation of PV cells in the medial septum (MS) at 40 Hz in these mice improves performance in an object place recognition task. However, this does not show that slow gamma oscillations are essential, as the authors did not test the inhibition of slow gamma in wild-type mice, and many other changes in neural function in the J20 mice might also explain impaired performance. The results are also far from establishing a causal link between slow gamma and spatial memory, because optogenetic stimulation of MS-PV cells have other effects than just enhancing hippocampal gamma.

2. In several places the authors mistakenly conclude that there is no effect based on the lack of statistically significant effect. It may be appropriate to remind the authors that 'absence of evidence is not evidence of absence'. For example, when the authors compare 40 Hz stimulation during the sample and retrieval phase of the task, they base their conclusion on a significant effect during the retrieval phase (0.66 ± 0.05 , $n = 7$) contrasted with an absence of significant effect of stimulation during the sample phase (0.59 ± 0.06 , $n = 5$). However, just because 0.59 is not significantly different from 0.50, it does not mean that there is no effect. Importantly, the authors used a smaller n in the latter case, and it would be necessary to report the statistical power with $n = 5$, assuming an effect size of the same magnitude as for the stimulation during the retrieval phase. Moreover, I believe the hypothesis that should be tested here is that there is a difference in effect depending on whether the 40 Hz stimulation was delivered during sample or retrieval phase, i.e. a two-sample test. There does not seem to be a significant difference between these conditions in Figure 4n, and the authors should report the results accordingly, or, preferably, increase the n number for stimulation during the sample phase.

3. There is also a problem with the reported frequency selectivity. The authors compare 40 Hz and 80 Hz stimulation of MS-PV cells, but, according to the authors, the 80 Hz stimulation did not translate into 80 Hz LFP in the hippocampus (Figures 2m and 3h,i). Since they could not successfully drive fast hippocampal gamma in their experiments, it does not make much sense to me to conclude that only slow hippocampal gamma affects task performance.

MINOR POINTS:

1. The authors argue in the Introduction that memory retrieval is supported by slow gamma recorded in the stratum radiatum of CA1. Yet, in this manuscript, the authors record slow gamma in stratum lacunosum-moleculare. The authors need to give a rationale or explanation for this choice, and discuss the possible interpretational implications.

2. There are some details in the abstract that should be clarified or corrected:

a. Line 15: It is not clear what 'slow gamma phase amplitude coupling' refers to. Do they mean 'theta-gamma phase-amplitude coupling'?

b. Line 19-20: '40 Hz (but not other frequencies) restores hippocampal slow gamma oscillations'. I think the authors tested only one other frequency, so this should read '40 Hz, but not 80 Hz, restores hippocampal gamma oscillations'.

3. There are some details in the figures that should be corrected:

- a. In Figure 1, scale bar is missing in panel d; and vertical scale bars are missing in panels e and f.
- b. In Figure 2, scale bar is missing in panel b.
- c. In Supplemental figure 10, the authors use 'Log frequency (Hz)' as x-axis label. They have actually plotted 'Frequency (Hz)' but on a log scale.

4. There are numerous grammatical errors, misspellings and typos throughout the manuscript, and the reference list does not seem to have been proof read, as there are inconsistencies regarding first letter capitalization or not in titles (e.g. compare ref. 1 and 2), some journal names are missing (e.g. ref. 7, 10 and 33), and journals are referred to inconsistently (see e.g. ref. 44 and 45; 63 and 64).

Reviewer #2 (Remarks to the Author):

In this revision, the authors have included substantial additional characterization of the electrophysiological and behavior effects of driving slow gamma in hippocampus via medial septum PV cell stimulation. They have substantially rewritten the manuscript including more discussion of the relevant literature and more technical descriptions of their methods. They have also improved the clarity of their figures. Overall, I find the manuscript to be vastly improved. A few issues remain, however, described below.

Major points:

The authors added key behavioral controls during the novel object place recognition task, as well as introducing two new control groups including 80 Hz stimulation during the test phase and 40 Hz stimulation during the encoding phase. The results of the behavioral assay are very interesting, but the authors were unable to replicate this result in another task. Furthermore, while they now include an 80 Hz stimulation group, they did not include a slower stimulation group, like 20 Hz, which would show that the stimulation must be slow gamma specifically. As a result the claim of "reinstatement of memory recall" still rests on a single group of 7 mice and that this stimulation must be 40 Hz, rests on another single group of 5 mice. Therefore, I recommend toning down and qualifying these claims. For example the title of the paper is "Optogenetic gamma stimulation rescues memory retrieval in Alzheimer's disease mouse model" but this is not the most robust result of the manuscript AND it leaves out many other interesting results, like the slow gamma deficits they find in J20 mice and that "optogenetic stimulation of parvalbumin neurons at 40 Hz (but not other frequencies) restores hippocampal slow gamma oscillations power and phase-amplitude coupling of the J20 AD mouse model," as the authors state in the abstract. In addition, the authors should qualify these claims in the discussion.

The authors state that they were unable to train J20 mice to perform an alternation task to a 30s delay and therefore did not replicate their stimulation results with this task. However, that seems an excellent opportunity to see if stimulation would allow these mice to learn the task. Why not try it? I don't understand the logic here. As described above, improved performance following stimulation in another task would go far to support the claim of memory retrieval reinstatement.

Furthermore, in figure 4n, the PVJ20+ ChETA, 40Hz stim during sample (orange) group includes 5 animals. Four of these animals perform above chance at similar levels to the PVJ20- group (black) and one animal that performs well below chance that could be an outlier. If this animal is an outlier, it suggests that stimulation during sample also improves performance. Further clarification of the results in this group is needed.

Minor points:

- Figure 1p and 1q – specific differences are indicated but cannot be seen in the current figure. Please stretch the x-axis to make this difference obvious.
- All color codes should be explicitly stated in the figure caption, not just shown in the figure.
- The authors claim “optogenetic stimulation of parvalbumin neurons at 40 Hz (but not other frequencies) restores hippocampal slow gamma oscillations power and phase-amplitude coupling.” This is a very interesting finding. The authors show that gamma stimulation of PV+ medial septum cells increases slow gamma power and theta-slow gamma coupling in J20+ mice compared to no stimulation. How does this induced activity compare to endogenous activity in J20- mice? Does stimulation restore these levels back to that of healthy mice? Boost levels above healthy mice? Something else?
- Figure 4n – Is PVJ20- YFP (black) statistically different from PVJ20+ ChETA, 40Hz stimulation during test (blue)?
- Some typos remain

Reviewer #3 (Remarks to the Author):

First of all, I want to congratulate the authors for their work, the manuscript clearly improved. While I have absolutely no doubt about the effect of the optogenetic stimulation on memory performances of the J20 mice, I still have hard time with the electrophysiological analysis. Here are more specific comments:

- 1) For the data presented in the figure 1, what is the physiological state of the animal? I guess that the recordings were performed during home-cage behavior, but when exactly (active wake???)
- 2) From my understanding of the figure 1, there is a dramatic decrease in gamma power, both in slow and fast gamma (clear in fig 1j and 1o). What I do not understand, is how the figure 1r and 1s were done. Given the huge decrease in power, what does a z-score on nothing mean? I would actually prefer to see the raw power values. Further, the suppl figure 3e clearly indicate that MI is not altered in Tg mice. As such, I'm really wondering why the authors spend so much time on coupling.... For me, the main results is that restoring gamma oscillations in the hippocampus is sufficient to improve memory. By itself, this is already a very important results.
- 3) In the discussion, the authors state that they established for the first time a link between slow gamma oscillations and spatial memory. But what about the recent article from Martorell et al., who showed that GENU8 increase memory in AD mice ?

In summary, as for the 1st round of review, I like the study, but the electrophysiological analysis is not convincing enough. I think that over-processing the results (z-score...) hide the fundamental results that lie in the increase of gamma power (it seems both at the slow and fast gamma range).

- 4) The authors spent a great deal of the introduction and discussion talking about the MSPV cells and desinhibition of the hippocampus. This is justified and should be keep in the manuscript. What the authors never mentioned, is the effect of the optogenetic stimulation on other MS neuronal population, as neurons in the MS are all connected (Leao et al., 2015). An interesting idea would be that activation of MSPV cells rhythmically drive cholinergique and GABAergic MS cells....

Reviewers' comments:

Reviewer #1 (Remarks to the Author):

The revised manuscript is in large parts rewritten and much improved.

However, I still have some concerns about the presentation. Most importantly, the authors need to tone down several of their conclusions, which are currently not supported by the data.

1. The authors suggest that 'slow gamma oscillations are essential for memory retrieval' (lines 23-24) and claim that their study 'establishes a causal link between hippocampal slow gamma oscillations and spatial memory' (lines 219-220). Neither of these statements is correct. The authors show that hippocampal slow gamma, as recorded in the stratum lacunosum-moleculare, has lower power in the J20 mouse model of Alzheimer's disease and that optogenetic stimulation of PV cells in the medial septum (MS) at 40 Hz in these mice improves performance in an object place recognition task. However, this does not show that slow gamma oscillations are essential, as the authors did not test the inhibition of slow gamma in wild-type mice, and many other changes in neural function in the J20 mice might also explain impaired performance. The results are also far from establishing a causal link between slow gamma and spatial memory, because optogenetic stimulation of MS-PV cells have other effects than just enhancing hippocampal gamma.

We thank reviewer #1 for this suggestion and acknowledge that the extend/limits of our demonstrations have to be clearly established and discussed in the manuscript. In particular, the question of causality is an important one and we agree that the angle we propose in this manuscript is rather 'in support of' than a 'definitive' demonstration of causality. We have removed the sentence '[...] establishes a causal link between hippocampal slow gamma oscillations and spatial memory' from the manuscript, highlight the electrophysiological data instead, and conclude with 'Our results suggest that more direct neural network manipulations could be employed as novel therapeutics to reverse memory loss in AD, and supports the role of hippocampal slow gamma oscillations in spatial memory.

' (lines 304-306).

2. In several places the authors mistakenly conclude that there is no effect based on the lack of statistically significant effect. It may be appropriate to remind the authors that 'absence of evidence is not evidence of absence'. For example, when the authors compare 40 Hz stimulation during the sample and retrieval phase of the task, they base their conclusion on a significant effect during the retrieval phase (0.66 +/- 0.05, n = 7) contrasted with an absence of significant effect of stimulation during the sample phase (0.59 +/- 0.06, n = 5). However, just because 0.59 is not significantly different from 0.50, it does not mean that there is no effect. Importantly, the authors used a smaller n in the latter case, and it would be necessary to report the statistical power with n = 5, assuming an effect size of the same magnitude as for

the stimulation during the retrieval phase. Moreover, I believe the hypothesis that should be tested here is that there is a difference in effect depending on whether the 40 Hz stimulation was delivered during sample or retrieval phase, i.e. a two-sample test. There does not seem to be a significant difference between these conditions in Figure 4n, and the authors should report the results accordingly, or, preferably, increase the n number for stimulation during the sample phase.

We agree with reviewer #1 that definitive conclusions cannot be made based on lack of statistical significance. Importantly, to clarify the analysis performed here, we did not compare groups with 0.5 values (one-sample t-tests) as it was done in the first version of the manuscript, but used 2-way repeated-measures ANOVA. Since more than two groups are being tested over two days, this statistical approach is more appropriate than multiple t-tests. As reviewer #1 pointed out, while the group receiving stimulation during the encoding phase is not statistically different from the PVJ20+ YFP group, we also found no statistical difference between the encoding and the retrieval 40 Hz stimulation group. We now detail these analyses/results in the text (lines 238-241) and modified our conclusions accordingly (lines 271-274). We also calculated the effect size and found a partial η^2 (most widely used for repeated measures ANOVA) value of 0.26 (now added to the results, line 235, and detailed in the method section, lines 683-685), and a partial ω^2 (less used but more stringent than partial η^2) value of 0.18 (not included in the manuscript) which are both considered large effects. We now added partial η^2 to the manuscript.

3. There is also a problem with the reported frequency selectivity. The authors compare 40 Hz and 80 Hz stimulation of MS-PV cells, but, according to the authors, the 80 Hz stimulation did not translate into 80 Hz LFP in the hippocampus (Figures 2m and 3h,i). Since they could not successfully drive fast hippocampal gamma in their experiments, it does not make much sense to me to conclude that only slow hippocampal gamma affects task performance.

We agree with reviewer #1 that 80 Hz stimulation have to be justified more thoroughly. We apologize for the lack of clarity when introducing this experiment. Importantly, PV neurons are able to follow 80 Hz stimulation (fig. 2f-h), and we are able to drive hippocampal rhythms at 80 Hz when stimulating at that frequency (fig. 2j,n; supplementary fig. 6h; supplementary fig. 8c; supplementary fig. 10j). In spite of our ability to pace hippocampal oscillations at 80 Hz, doing so did not increase fast gamma phase-amplitude coupling to theta phase (fig. 3g). This could be due to the fact that fast gamma coupling is preserved in PVJ20+ mice (fig. 1m, supplementary fig 3h) which could explain why increase in fast gamma band power are more difficult to induce in comparison to slow gamma band power. Importantly, while 80 Hz stimulation can pace MSPV cells, and induce 80 Hz rhythms in the hippocampus, they were not associated with significant increase in memory performance in the novel place object recognition task. We have now clarified these results (lines 172-177 & 241-243). We also now include a summary table (supplementary table 2) to facilitate the access to these key results.

MINOR POINTS:

1. *The authors argue in the Introduction that memory retrieval is supported by slow gamma recorded in the stratum radiatum of CA1. Yet, in this manuscript, the authors record slow gamma in stratum lacunosum-moleculare. The authors need to give a rationale or explanation for this choice, and discuss the possible interpretational implications.*

We acknowledge that while slow gamma has been suggested to originate from stratum radiatum, it can be recorded at most hippocampal layers. We concentrated our analysis in the lacunosum moleculare because we found most decreased slow gamma oscillations there (supplementary fig. 3). Moreover, since theta amplitude is largest closer to the hippocampal fissure and thus in the Im, and our measurements of gamma oscillation are normalized by theta power, analysing gamma power/coupling increased consistency in the results. We now justify this recording configuration more clearly in the manuscript (lines 114-118)

2. *There are some details in the abstract that should be clarified or corrected:*

a. *Line 15: It is not clear what 'slow gamma phase amplitude coupling' refers to. Do they mean 'theta-gamma phase-amplitude coupling'?*

This has been clarified and replaced as 'slow gamma phase-amplitude coupling to theta phase' in the abstract and throughout the manuscript.

b. *Line 19-20: '40 Hz (but not other frequencies) restores hippocampal slow gamma oscillations'. I think the authors tested only one other frequency, so this should read '40 Hz, but not 80 Hz, restores hippocampal gamma oscillations'.*

We agree that this should be clarified. We now state '[...] 40 Hz (but not 80 Hz), restores hippocampal slow gamma oscillations [...]' in the abstract

3. *There are some details in the figures that should be corrected:*

a. *In Figure 1, scale bar is missing in panel d; and vertical scale bars are missing in panels e and f.*

We have now added scale bars in these panels that have now been transferred in supplementary fig. 3.

b. *In Figure 2, scale bar is missing in panel b.*

We have now added a scale bar in panel b

c. *In Supplemental figure 10, the authors use 'Log frequency (Hz)' as x-axis label. They have actually plotted 'Frequency (Hz)' but on a log scale.*

We acknowledge that values themselves are not log, but the scale is. We corrected that mistake in the new version.

4. *There are numerous grammatical errors, misspellings and typos throughout the manuscript, and the reference list does not seem to have been proof read, as there are inconsistencies regarding first letter capitalization or not in titles (e.g. compare ref. 1 and 2),*

some journal names are missing (e.g. ref. 7, 10 and 33), and journals are referred to inconsistently (see e.g. ref. 44 and 45; 63 and 64).

Although we have proofread the manuscript several times by English native speakers, we apologize for typos that have been missed and done our best to have additional rounds of corrections. We thank for review #1 to point out these omissions.

- Capitalization has been harmonized to sentence casing for all references
- Omitted journal names have been added

Reviewer #2 (Remarks to the Author):

In this revision, the authors have included substantial additional characterization of the electrophysiological and behavior effects of driving slow gamma in hippocampus via medial septum PV cell stimulation. They have substantially rewritten the manuscript including more discussion of the relevant literature and more technical descriptions of their methods. They have also improved the clarity of their figures. Overall, I find the manuscript to be vastly improved. A few issues remain, however, described below.

Major points:

The authors added key behavioral controls during the novel object place recognition task, as well as introducing two new control groups including 80 Hz stimulation during the test phase and 40 Hz stimulation during the encoding phase. The results of the behavioral assay are very interesting, but the authors were unable to replicate this result in another task. Furthermore, while they now include an 80 Hz stimulation group, they did not include a slower stimulation group, like 20 Hz, which would show that the stimulation must be slow gamma specifically. As a result the claim of “reinstatement of memory recall” still rests on a single group of 7 mice and that this stimulation must be 40 Hz, rests on another single group of 5 mice. Therefore, I recommend toning down and qualifying these claims. For example the title of the paper is “Optogenetic gamma stimulation rescues memory retrieval in Alzheimer’s disease mouse model” but this is not the most robust result of the manuscript AND it leaves out many other interesting results, like the slow gamma deficits they find in J20 mice and that “optogenetic stimulation of parvalbumin neurons at 40 Hz (but not other frequencies) restores hippocampal slow gamma oscillations power and phase-amplitude coupling of the J20 AD mouse model,” as the authors state in the abstract. In addition, the authors should qualify these claims in the discussion.

We thank reviewer #2 for the additional review and suggestions. We have reformulated our claims and conclusions based on the results showed here. We now state in the abstract '[...] 40 Hz (but not 80 Hz), restores hippocampal slow gamma oscillations [...]'. We now report the effect size for our repeated measure ANOVA on behavioral data and detail all post-hoc analyses (line 212).

The authors state that they were unable to train J20 mice to perform an alternation task to a 30s delay and therefore did not replicate their stimulation results with this task. However,

that seems an excellent opportunity to see if stimulation would allow these mice to learn the task. Why not try it? I don't understand the logic here. As described above, improved performance following stimulation in another task would go far to support the claim of memory retrieval reinstatement.

We agree with reviewer #2 that additional behavioral data could support claims of memory retrieval. Since PVJ20+ mice did not learn the task in our conditions, it is hard to establish whether this was due to encoding/retrieval/working memory defects, or simply that the non-match-to-place alternation rule has not been learned by these mice. While these experiments could indeed provide valuable insights, we hope reviewer #2 understands that these experiments were beyond the scope of what was currently feasible for this manuscript. We are planning on following up on these studies with more dedicated behavioral tests in the future.

Furthermore, in figure 4n, the PVJ20+ ChETA, 40Hz stim during sample (orange) group includes 5 animals. Four of these animals perform above chance at similar levels to the PVJ20- group (black) and one animal that performs well below chance that could be an outlier. If this animal is an outlier, it suggests that stimulation during sample also improves performance. Further clarification of the results in this group is needed.

We agree with reviewer #2 (and as mentioned with reviewer #1) that analyses for this experiment have to be more detailed for each group. We now extended our post-hoc analyses for figure 4n. While the group receiving stimulation during the encoding phase is not statistically different from the PVJ20+ YFP group, we also found no statistical difference between the encoding and the retrieval 40 Hz stimulation group. We now detail these analyses in the text (lines 238-241) and modified our conclusions accordingly (lines 270-274).

Minor points:

- *Figure 1p and 1q – specific differences are indicated but cannot be seen in the current figure. Please stretch the x-axis to make this difference obvious.*

The new version of the manuscript now displays these panels (now labeled 1j,k) with stretched x-axes.

- *All color codes should be explicitly stated in the figure caption, not just shown in the figure.*

Color codes have now been stated in every figure captions when applicable.

- *The authors claim “optogenetic stimulation of parvalbumin neurons at 40 Hz (but not other frequencies) restores hippocampal slow gamma oscillations power and phase-amplitude coupling.” This is a very interesting finding. The authors show that gamma stimulation of PV+ medial septum cells increases slow gamma power and theta-slow gamma coupling in J20+ mice compared to no stimulation. How does this induced activity compare to*

endogenous activity in J20- mice? Does stimulation restore these levels back to that of healthy mice? Boost levels above healthy mice? Something else?

This question is indeed relevant for the interpretation of the results. We now added this analysis in figure 3j, and compared PVJ20+ gamma oscillation with PVJ20- controls, before and during 40 Hz stimulation. We found that before 40 Hz optogenetic stimulation, slow gamma power in PVJ20+ is significantly reduced (2ANOVA, $F_{(1, 480)} = 49.66$, $p < 0.0001$) (as shown earlier in the manuscript) while during stimulation both groups did not differ significantly (2ANOVA, $F_{(1, 480)} = 1.231$, $p = 0.2678$) and the increase of slow gamma power before and during 40 Hz stimulation in PVJ20+ mice was significant (RMANOVA, $F_{(1, 120)} = 317.0$, $p < 0.0001$), suggesting that during 40 Hz optogenetic stimulation, phase-amplitude coupling of slow gamma oscillations is restored to normal levels. These results are now included (lines 201-205).

- *Figure 4n – Is PVJ20- YFP (black) statistically different from PVJ20+ ChETA, 40Hz stimulation during test (blue)?*

As mentioned earlier, we have now added post-hoc results for this comparison and while 40 Hz-encoding stimulation group is not statistically different from the PVJ20+ YFP group, we also found no statistical difference between the encoding and the retrieval 40 Hz stimulation groups (lines 238-241). We have now included these results in our discussion (lines 270-274).

- *Some typos remain*

We thank reviewer #3 for all aforementioned suggestions and apologize for the remaining typos. We have carefully proofread our manuscript and corrected all the typos we found.

Reviewer #3 (Remarks to the Author):

First of all, I want to congratulate the authors for their work, the manuscript clearly improved.

While I have absolutely no doubt about the effect of the optogenetic stimulation on memory performances of the J20 mice, I still have hard time with the electrophysiological analysis.

Here are more specific comments:

1) *For the data presented in the figure 1, what is the physiological state of the animal? I guess that the recordings were performed during home-cage behavior, but when exactly (active wake???)*.

We thank reviewer #3 for the encouraging comments and we apologize for the lack of clarity of this section. For data presented in fig. 1, mice are actively exploring a circular platform. In addition to fig. 1b showing the recorded state, we now added in the text that mice are actively exploring the platform (lines 100-101). Importantly, because mice are either stationary or running at different speed (which strongly correlates with theta and gamma power), we normalized gamma power to corresponding theta power/values so as to control for the high variability in states (this is clarified lines 125-126 and see further responses below).

2) *From my understanding of the figure 1, there is a dramatic decrease in gamma power, both in slow and fast gamma (clear in fig 1j and 1o). What I do not understand, is how the figure 1r and 1s were done. Given the huge decrease in power, what does a z-score on nothing mean? I would actually prefer to see the raw power values. Further, the suppl figure 3e clearly indicate that MI is not altered in Tg mice.*

As such, I'm really wondering why the authors spend so much time on coupling.... For me, the main results is that restoring gamma oscillations in the hippocampus is sufficient to improve memory. By itself, this is already a very important results.

We apologize for being unclear. In figure 1l,m (previously r and 1s), slow and fast gamma are indeed z-scored, meaning that the only conclusion that can be drawn is on the amount of phase modulation of gamma amplitude, regardless of potential amplitude differences. This is a classical approach that has been used by most studies of phase-amplitude coupling (Belluscio et al., cited; Schomburg et al., cited; Mably et al., cited; Zheng et al., cited). Importantly, slow and fast gamma oscillations are reduced, but not null (fig. 1d,j,k), so z-scoring can still be performed in our conditions. We agree that the most important result is the reduction of gamma power/amplitude, however we find potential comparisons with previous studies of coupling relevant to this manuscript.

Secondly, as pointed by reviewer #3, fast gamma MI is unaltered in PVJ20+ mice (which is also shown in fig. 1m). However, slow gamma MI is greatly reduced in supplementary fig. 3. We apologize for the lack of visibility and added an arrow and corresponding legend to guide the reader's attention to that result. Additionally, we have now included a supplementary table (supplementary table 2) that summarizes all the neurophysiological phenotype of PVJ20+ mice, as well as the effects of optogenetic stimulation.

3) *In the discussion, the authors state that they established for the first time a link between slow gamma oscillations and spatial memory. But what about the recent article from Martorell et al., who showed that GENUS increase memory in AD mice ?*

We agree with reviewer #3 and acknowledge that previous studies including (Iaccarino et al., 2016; cited) and (Martorell et al., 2019; now cited) have already used gamma range stimulation in the context of AD pathology. The study of Martorell et al. is particularly interesting in that context since they used a similar behavioral test but the treatment was much longer (5 days chronic). We now correct that this is the first optogenetic study including acute behavioral effects (lines 74-76). We now discuss these results (lines 253-255)

In summary, as for the 1st round of review, I like the study, but the electrophysiological analysis is not convincing enough. I think that over-processing the results (z-score...) hide the fundamental results that lie in the increase of gamma power (it seems both at the slow and fast gamma range).

We apologize for the lack of clarity in the presentation of results. Two distinct elements are being analyzed in our study: gamma power, and coupling of gamma to theta phase. Z-scoring has been widely used in the context of phase amplitude coupling (Belluscio et al., cited;

Schomburg et al., cited; Mably et al., cited; Zheng et al., cited), and we found that comparing similar behavioral states (and thus comparable theta/gamma conditions) was challenging without normalizing data to theta power. We have attempted to normalize gamma power directly to the velocity of mice, which led to more variability since velocity values have to be interpolated from video camera data. Importantly, we could pace hippocampal oscillations in both slow (40 Hz) and fast (80 Hz) gamma ranges, but only slow gamma optogenetic stimulation were efficient at restoring slow gamma phase-amplitude coupling to theta and spatial memory. All these results have now been summarized more clearly in supplementary table 2.

4) *The authors spent a great deal of the introduction and discussion talking about the MSPV cells and desinhibition of the hippocampus. This is justified and should be kept in the manuscript. What the authors never mentioned, is the effect of the optogenetic stimulation on other MS neuronal population, as neurons in the MS are all connected (Leao et al., 2015). An interesting idea would be that activation of MSPV cells rhythmically drive cholinergic and GABAergic MS cells....*

This is a very interesting point. It is very likely that MSPV cells rhythmically drive other GABAergic cells within the medial septum, and as mentioned by reviewer #3, Leao et al. (2015; now cited) also show that ~10% of GABAergic cells project onto cholinergic cells, so cholinergic effects cannot be entirely excluded. We now discuss this point (lines 287-289).

REVIEWERS' COMMENTS:

Reviewer #1 (Remarks to the Author):

The authors have done a great job in revising their manuscript, and I have no further suggestions for improvement.

Reviewer #2 (Remarks to the Author):

The authors have satisfied almost all of my concerns. My remaining comment is that the authors should further tone down their claims that encoding is intact in these J20 mice as they did not find a significant difference between 40Hz stimulation during encoding and retrieval phases of the task.

Reviewer #3 (Remarks to the Author):

The current version of the manuscript is again improved compared to the previous one. I acknowledge the work done by the authors (e.g. The discussion is way more cautious now, and more related to the actual results of the present study).

However, I still have concerns about the interpretation of the results, based on the data provided in the manuscript.

The authors linked hippocampal slow gamma to memory retrieval as the basis for their optogenetic experiment. I think that this "shortcut" (as stated in the discussion with great caution) is somehow misleading. Indeed, in the seminal paper by Colgin et al and the follow-ups from either the Buzsaki's group or Dupret group, SG is just a "communication channel" between CA3 and CA1. This is the information convey by CA3 (using SG as a carrier) which is important for memory retrieval. In the results provided by the authors, their optogenetic stimulation do not increase current loading in the Str.radiatum (Using CSD, see supp figure 10). As such, they, in my opinion, do not talk about the same process as the one reported in the aforementioned studies (as the authors stated, "the recording conditions do not permit independent component analysis (ICA) or other decomposition methods that would allow us to identify gamma generators within the hippocampus").

Further, experimental data with optogenetic just reinforce this claim as the 40Hz stimulation during the sample phase seems also to have an effect on memory performances (the number of animals is definitively too low, and, as noted by other reviewer, with 1 outlier).

To conclude, I think that the authors clearly made their point that restoring slow gamma dynamic in hippocampal networks have a beneficial effect on memory processes in their animal model of AD (being during the sample or test phase). I don't think however that they can separate encoding and retrieval processes with their results. As such, they did not "supports the role of hippocampal slow gamma oscillations in spatial memory retrieval".

I do not ask the authors to perform additional experiments, and I believe that the results are important enough to deserve publication in Nature communication. However, I would extensively rewrite the introduction to be more in line with the discussion (by toning down the part on SG and FG and associated cognitive process which can be misleading) and focus on the main results, which is that restoring slow-gamma dynamics in hippocampal networks alleviate memory deficits in their animal model of AD.

REVIEWERS' COMMENTS:

Reviewer #1 (Remarks to the Author):

The authors have done a great job in revising their manuscript, and I have no further suggestions for improvement.

We thank reviewer #1 for the time spent on the manuscript.

Reviewer #2 (Remarks to the Author):

The authors have satisfied almost all of my concerns. My remaining comment is that the authors should further tone down their claims that encoding is intact in these J20 mice as they did not find a significant difference between 40Hz stimulation during encoding and retrieval phases of the task.

We agree that further behavioral testing could be performed to specifically separate effects of optogenetic stimulation during encoding and retrieval. We acknowledge this perspective in our discussion. We thank reviewer #1 for the time spent on the manuscript

Reviewer #3 (Remarks to the Author):

The current version of the manuscript is again improved compared to the previous one. I acknowledge the work done by the authors (e.g. The discussion is way more cautious now, and more related to the actual results of the present study).

However, I still have concerns about the interpretation of the results, based on the data provided in the manuscript.

The authors linked hippocampal slow gamma to memory retrieval as the basis for their optogenetic experiment. I think that this “shortcut” (as stated in the discussion with great caution) is somehow misleading. Indeed, in the seminal paper by Colgin et al and the follow-ups from either the Buzsaki’s group or Dupret group, SG is just a “communication channel” between CA3 and CA1. This is the information convey by CA3 (using SG as a carrier) which is important for memory retrieval. In the results provided by the authors, their optogenetic stimulation do not increase current loading in the Str.radiatum (Using CSD, see supp figure 10). As such, they, in my opinion, do not talk about the same process as the one reported in the aforementioned studies (as the authors stated, “the recording conditions do not permit independent component analysis (ICA) or other decomposition methods that would allow us to identify gamma generators within the hippocampus”).

Further, experimental data with optogenetic just reinforce this claim as the 40Hz stimulation during the sample phase seems also to have an effect on memory performances (the number of animals is definitively too low, and, as noted by other reviewer, with 1 outlier).

To conclude, I think that the authors clearly made their point that restoring slow gamma dynamic in hippocampal networks have a beneficial effect on memory processes in their animal model of AD (being during the sample or test phase). I don’t think however that they can separate encoding and retrieval processes with their results. As such, they did not “supports the role of hippocampal slow gamma oscillations in spatial memory retrieval”.

I do not ask the authors to perform additional experiments, and I believe that the results are important enough to deserve publication in Nature communication. However, I would extensively rewrite the introduction to be more in line with the discussion (by toning down the part on SG and FG and associated cognitive process which can be misleading) and focus on the main results, which is that restoring slow-gamma dynamics in hippocampal networks alleviate memory deficits in their animal model of AD.

We thank reviewer #3 for these interesting comments related to our previous revision. We agree that, as mentioned by reviewer #2, separating effects of optogenetic stimulation on encoding vs retrieval would require more behavioral data. Currently, while we are confident that 40 Hz stimulation are effective when restricted to retrieval of spatial memory, we cannot conclude much on the encoding group as they do not statistically differ from untreated, PVJ20+, YFP mice. Although it is clear that some mice in the 40Hz, encoding stim group performed well above chance, further studies would be required to confirm that this result is maintained. We have now changed the title, abstract, and text to reflect these results. On the other, hand, reviewer #3 raises an interesting point: in my understanding, the question is whether physiological slow gamma oscillations (and underlying slow gamma generators) are being driven in our conditions. While it is hard to completely answer this question, we are definitely 'adding 40 Hz oscillations' in hippocampal layers that do not normally display such oscillations, such as the stratum lacunosum moleculare. In that sense, our optogenetic stimulation elicits 'artificial' responses. However, this does not necessarily mean that physiological slow gamma is not being driven. In place of modifying the introduction, which we believe mentions important studies of slow and fast gamma rhythms that are essential to understand our study, we propose to add a discussion that reflects the point brought up by reviewer #3. We thank again reviewer #3 for the time spent reviewing this article and believe these comments contributed to strengthen this manuscript.